# HiBio-ST: A Hierarchical Multimodal Foundation Model with Biological Prior Anchors for Spatial Transcriptomics

## Abstract

Spatial transcriptomics (ST) enables medical computer vision researchers to uncover the molecular relationships underlying tissue morphology. However, most existing vision–omics models are built on limited and homogeneous datasets, rendering them task-specific and with poor generalizability. Recent multimodal foundation models attempt to bridge histology and gene expression via contrastive objectives; however, they fail to effectively model spot-specific molecular context and overlook spatial dependencies by treating each spot–patch pair in isolation. To bridge these gaps, we present HiBio-ST, a novel hierarchical multimodal foundation model guided by biological prior anchors for ST analysis. HiBio-ST employs a progressive multi-level alignment pretraining pipeline to harmonize visual context with molecular identities. A TF–IDF reweighting strategy is first applied to highlight spatially informative "keyword" genes within ST profiles, reducing the dominance of ubiquitous housekeeping signals. Curated pathway anchors are then incorporated to inject global biological knowledge into the representation space. Moreover, hierarchical region-aware clustering united contiguous meso-scale regions into coherent structural patterns, allowing the model to capture higher-order spatial organization. We evaluated HiBio-ST on four downstream tasks across multiple datasets. Experimental results demonstrate that HiBio-ST consistently achieves state-of-the-art performance, underscoring its broad applicability in spatial transcriptomics modeling.

## 1 Introduction

Spatial transcriptomics (ST) has emerged as a transformative technology that profiles gene expression within tissue architecture (Ståhl et al., 2016; Williams et al., 2022; Asp et al., 2020; Burgess, 2019). By providing a molecular atlas aligned with histological structures, ST has advanced our understanding including disease mechanisms and tumor microenvironmental heterogeneity (Arora et al., 2023; Smith et al., 2024; Jones et al., 2024; Dent et al., 2013), while the high cost and technical complexity of ST limit scalability in clinical and research settings (Zhu et al., 2025a; Piñeiro et al., 2022). Pathology slides, routinely acquired in clinical workflow, carry morphological cues correlated with underlying gene expression (Ash et al., 2021). This intrinsic relationship motivates cost-effective vision–omics models (He et al., 2020; Yang et al., 2023; Xie et al., 2023).

However, existing vision-driven methods remain task-specific and are trained on relatively limited, homogeneous datasets (He et al., 2020; Zeng et al., 2022), yielding representations that generalize poorly across tissues, disease types, or sequencing platforms (Pang et al., 2021; Zhu et al., 2025b; Shi et al., 2024). Their single-objective optimization further limits adaptability to new downstream tasks, making it difficult to extend beyond the scope of their original design. These limitations underscore the need for a foundation-model paradigm that offers transferable and robust representation for multiple applications ranging from gene expression prediction to biomarker discovery.

Recent multimodal foundation model attempts to bridge histology and gene profile via contrastive learning (Chen et al., 2025); however, two issues persist. First, the high-expression-only strategy overlooks spot-specific signals and instead favors ubiquitous housekeeping genes. Consequently, critical yet low-expressed markers like BRCA1 or FOXP3 are often missed in selected gene pro-

files, whereas those pervasive ones like ribosomal proteins dominate the representation. Thus, yielding nearly identical profiles across spots and hindering their effective distinction (Charitakis et al., 2023; Shang et al., 2025). Secondly, treating spot–patch pairs in isolation, the current method ignores spatial dependencies that underlie microenvironmental interactions and regional functional specialization, thus limiting their ability to faithfully capture multiscale biology.

To address these limitations, we propose HiBio-ST, a hierarchical multimodal foundation model with biological prior anchors tailored for spatial transcriptomics analysis. HiBio-ST leverages a staged pretraining strategy that progressively aligns histological features with molecular identities across multiple levels. Gene representations are reformulated through a novel TF–IDF scheme that suppresses housekeeping noise while highlighting spot-specific informative signals. This local patch-spot alignment is further reinforced by curated pathway anchors, which inject global biological constraints into the embedding space. At a broader scale, hierarchical region-aware clustering consolidates neighboring spots into coherent structural patterns, granting the model the capacity to recognize complex tissue organization. Together, HiBio-ST delivers a unified framework that integrates signals across multiple biological scales.

Our contributions can be summarized as three folds:

- We introduce a biologically informed alignment scheme that redefines gene representations through TF–IDF reweighting, highlighting spatially informative marker genes and anchoring them to curated pathways to infuse global functional semantics

- We develop a spatial-aware modeling strategy that hierarchically consolidates neighboring spots into coherent meso-scale structural units, thereby bridging fine-grained molecular cues with higher-level tissue patterns.

- We establish HiBio-ST as a multimodal foundation model through a progressive multi-level alignment pretraining pipeline, demonstrating strong transferability and robust generalization across datasets with superior performance on diverse downstream tasks.

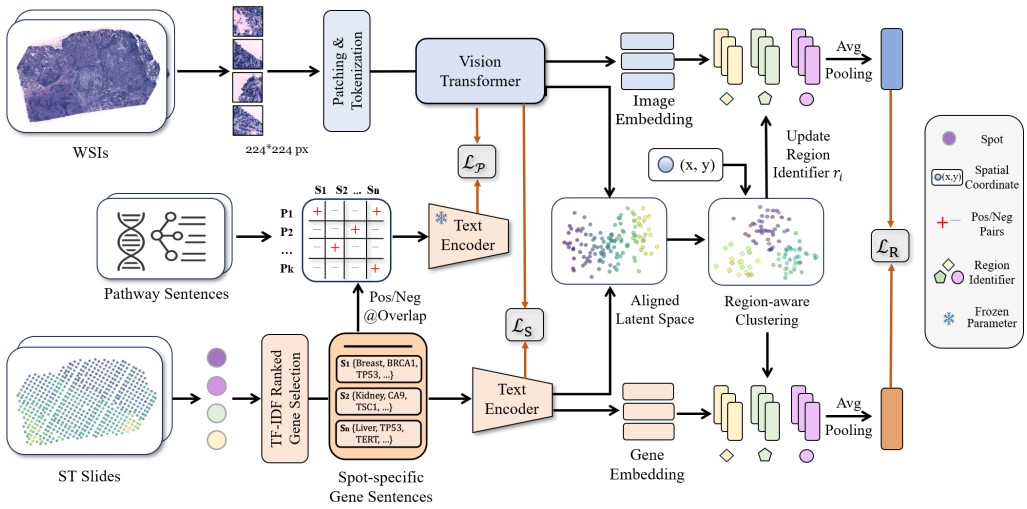

Figure 1: **Overview of HiBio-ST pretraining pipeline.** HiBio-ST adopts a progressive multi-level alignment strategy to harmonize visual features with molecular profiles. First, at the local scale, vision-omics pairs are constructed by selecting informative "keyword" genes through a TF-IDF reweighting scheme to support fine-grained alignment. Then, curated pathway anchors are matched to each spot by computing gene-set overlaps, thereby defining positive and negative pairs at the functional scale, which serve as soft labels to inject biological knowledge. After initial pretraining at the spot-patch level, spatially aware clustering assigns region identifiers to contiguous spots, thereby integrating local patches into region-aware representations. Thus, HiBio-ST progressively incorporates local molecular signals and vision features with pathway-level semantics and meso-scale structures, achieving a unified multimodal representation across multiple sources and scales.

## 2 METHOD

Our HiBio-ST adopts a progressive multi-level pretraining pipeline, as presented in Figure 1. At the local scale, patch–spot pairs are constructed by applying TF–IDF weighting to extract informative "keyword" genes that support fine-grained alignment. At the functional scale, overlaps between pathway gene sets and spot sentences act as soft labels to inject biological knowledge into representation space. Spatial-aware clustering aggregates neighboring spots into coherent regions, yielding region-aware representations. In this way, HiBio-ST integrates local molecular signals and histological features with pathway-level semantics and meso-scale structures.

### 2.1 TF–IDF REWEIGHTED GENE SELECTION

Gene sentence selection enables informative modeling of molecular context. However, a common practice of selecting top-expressed genes often overrepresents housekeeping genes, which are constitutively active but biologically uninformative for distinguishing cell types or spatial identities. Thus, we construct a gene sentence for each spot using a TF-IDF weighting scheme, which functions much like highlighting rare yet informative "keywords" in a document, thereby emphasizing context-specific molecular identity.

Before training, we operate on data in batches. Let $X \in \mathbb{R}^{B \times N_h}$ denote the expression matrix of a training batch containing $B$ spots and $N_h$ genes, where $X_{j,i}$ represents the raw expression value of gene $g_i$ in spot $s_j$. To reduce the influence of extreme values, we apply a log-transformation and define the term frequency (TF) as:

$$\text{TF}(g_i, s_j) = \frac{\log(1 + X_{j,i})}{\sum_{k=1}^{N_h} \log(1 + X_{j,k}) + \epsilon},$$

(1)

where $\epsilon$ is a small constant added to prevent division by zero. To capture each gene's global rarity, we precompute inverse document frequency (IDF) values over the full corpus $X_{\text{all}}$ as follows:

$$\text{IDF}(g_i) = \log\left(\frac{1 + M}{1 + \sum_{j=1}^{M} \mathbf{1}[X_{\text{all},j,i} > 0]}\right),$$

(2)

where $\mathbf{1}[\cdot]$ is the indicator function and $M$ refers to the total spot number. Since zero-filled entries satisfy $X_{j,i} = 0$, they are excluded from the IDF statistics, preserving the reliability of the global frequency estimate. The final TF–IDF weight for gene $g_i$ in spot $s_j$ is then computed as:

$$w_{j,i} = \text{TF}(g_i, s_j) \cdot \text{IDF}(g_i).$$

(3)

We then sort each row of the weight matrix $w_{j,1}, \ldots, w_{j,N_h}$ and extract the top-$K$ genes to form the gene sentence for spot $s_j$:

$$\mathcal{S}(s_j) = [g_{(1)}, g_{(2)}, \ldots, g_{(K)}],$$

(4)

where $g_{(i)}$ denotes the $i$-th highest-ranked gene based on TF–IDF score.Thus, the TF-IDF strategy combines local expression salience with global distinctiveness, yielding sparse yet biologically informative gene sentences that provide strong supervision for downstream multimodal learning.

### 2.2 PATHWAY-GUIDED BIOLOGICAL PRIOR INTEGRATION

To complement the initial spot-patch alignment and inject structured biological knowledge, we treat curated KEGG pathways (Kanehisa et al., 2023) as functional anchors. $G_s$ refers to the spot-specific gene set selected by the TF-IDF-based filtering strategy. Each pathway $P$ is associated with a gene set $G_P \subseteq \mathcal{G}$, describing a coherent biological function. To quantify the semantic alignment between spot $S$ and pathway $P$, we define a thresholded normalized overlap score:

$$\omega(P, S) = \delta\left(\frac{|G_P \cap G_S|}{\min(|G_P|, |G_S|)} > \tau\right) \cdot \frac{|G_P \cap G_S|}{\min(|G_P|, |G_S|)},$$

(5)

where $\delta(\cdot)$ denotes the indicator function, and $\tau$ is a predefined threshold (e.g., $\tau = 0.2$) to filter out low-relevance pathway–patch pairs. The score $\omega(P, S)$ serves directly as a soft supervision weight

in contrastive learning. Those pathway–patch pairs with $\omega(P, S) > 0$ are treated as positives with varying importance, while others are considered as negative pairs.

Each pathway is embedded as a prototype vector by averaging the embeddings of its member genes:

$$\mathbf{z}_P = \frac{1}{|G_P|} \sum_{g \in G_P} \mathbf{e}_g, \tag{6}$$

where $\mathbf{e}_g$ is the learned embedding of gene $g$. These pathway-level prototypes are used in tandem with spot-level gene representations to support multi-granularity alignment. By aligning histology features with these pathway prototypes under the soft-prior weighting, the model learns representations that capture both local expression patterns and higher-order functional semantics.

## 2.3 REGION-AWARE HIERARCHICAL REPRESENTATION

Existing supervision signals at the spot and pathway levels provide complementary perspectives on local expression patterns and global biological context. However, tissues function in contiguous meso-scale regions whose boundaries are not labeled. We therefore induce regions directly from the data by clustering spots with a distance that fuses spatial proximity and multimodal similarity, thus providing a spatial prior that stabilizes learning and reduces overfitting to purely local textures.

Concretely, we assign each spot $i$ a region identifier $r_i$ by joint representation. This clustering procedure is performed periodically during training to accommodate evolving representations. Each spot is represented by image embedding $\mathbf{v}_i$ extracted from ViT and text embedding $\mathbf{t}_i$ obtained from a text transformer. We encode each gene symbol as an atomic token to preserve its semantic granularity, whereas conventional subword tokenization fragments genes into partial components and undermines gene-level representation.

We then define the distance between two spots $i$ and $j$ as the fusion of spatial distance and extracted representations, denoted as:

$$D_{ij} = \alpha \cdot \frac{\|\mathbf{x}_i - \mathbf{x}_j\|_2 - d_{\min}}{d_{\max} - d_{\min}} + \beta \cdot \frac{1 - \cos([\mathbf{v}_i; \mathbf{t}_i], [\mathbf{v}_j; \mathbf{t}_j]) - c_{\min}}{c_{\max} - c_{\min}}, \tag{7}$$

where $\mathbf{x}_i \in \mathbb{R}^2$ is the spatial coordinate of spot $i$, and $[\mathbf{v}_i; \mathbf{t}_i]$ denotes the concatenated embedding of image and text features. The terms $d_{\min}, d_{\max}$ and $c_{\min}, c_{\max}$ denote the range of Euclidean and cosine distances across the dataset, ensuring both terms are normalized to $[0, 1]$. We set $\alpha = \beta = 0.5$.

A weighted $k$-nearest neighbor graph is then constructed with edge weights $W_{ij} = \exp(-D_{ij})$. Leiden clustering (Traag et al., 2019) is applied to maximize modularity:

$$Q = \frac{1}{2m} \sum_{i,j} \left[ W_{ij} - \frac{k_i k_j}{2m} \right] \cdot \mathbf{1}[r_i = r_j], \tag{8}$$

where $k_i = \sum_j W_{ij}$ is the weighted degree of node $i$, $m = \frac{1}{2} \sum_{i,j} W_{ij}$ is the total edge weight, and $\mathbf{1}[r_i = r_j]$ indicates whether nodes $i$ and $j$ are assigned to the same cluster. Consequently, each spot receives a region identifier $r_i$, thereby capturing spatially coherent structures in the tissue.

## 2.4 MULTI-LEVEL CONTRASTIVE ALIGNMENT

We adopt a hybrid contrastive loss to achieve multi-level alignment, including a spot-level loss $\mathcal{L}_s$, a pathway-guided soft contrastive term $\mathcal{L}_p$, and a regional hierarchical contrastive term $\mathcal{L}_r$. For each spot, it yields an image embedding $\mathbf{v}_i$ and a gene-sentence embedding $\mathbf{t}_i$. We enforce local correspondence with a symmetric InfoNCE loss. For the image-to-text direction, we have

$$\mathcal{L}_{\mathbf{v} \to \mathbf{t}} = -\frac{1}{B} \sum_{i=1}^{B} \log \frac{\exp(\mathbf{v}_i^\top \mathbf{t}i/\tau)}{\sum_{j=1}^{B} \exp(\mathbf{v}_i^\top \mathbf{t}_j/\tau)} \tag{9}$$

We define $\mathcal{L}_{\mathbf{t} \to \mathbf{v}}$ by swapping roles. The combined spot-level contrastive loss $\mathcal{L}_s$ is defined as:

$$\mathcal{L}_s = \frac{1}{2} \left( \mathcal{L}_{\mathbf{v} \to \mathbf{t}} + \mathcal{L}_{\mathbf{t} \to \mathbf{v}} \right) \tag{10}$$

which anchors patch-wise visual and transcriptomic features in a shared embedding space.

To inject structured knowledge, each histology embedding is mapped by the projection head and softly aligned with pathway prototypes $\mathbf{z}_{p_j}$ using normalized overlap weights $\omega_j$. These weights form a soft target distribution $q_i(j) \propto \exp(\omega_j)$, while $\hat{q}_i$ denotes the model-predicted probabilities obtained by computing dot-product similarity between the projected embedding and both positive and sampled negative prototypes, followed by a softmax normalization. The pathway-guided soft contrastive loss $\mathcal{L}_p$ is defined as

$$\mathcal{L}p = \frac{1}{B} \sum i = 1^B \left[ 1 - \exp\left( -\frac{1}{\alpha} \cdot \mathrm{KL}(\hat{q}_i \parallel q_i) \right) \right]. \tag{11}$$

After several rounds of uni-scale training, neighboring spots are clustered into regions $r$. We average member embeddings to form region prototypes, which summarize meso-scale tissue organization:

$$\bar{\mathbf{v}}_r = \frac{1}{|r|} \sum_{i \in r} \mathbf{v}_i, \quad \bar{\mathbf{t}}_r = \frac{1}{|r|} \sum_{i \in r} \mathbf{t}_i. \tag{12}$$

A symmetric InfoNCE loss is applied for further region-aware alignment:

$$\mathcal{L}_R = \frac{1}{2}\left( \mathcal{L}_{\bar{\mathbf{v}}_r \to \bar{\mathbf{t}}_r} + \mathcal{L}_{\bar{\mathbf{t}}_r \to \bar{\mathbf{v}}_r} \right), \tag{13}$$

This region-wise supervision encourages consistent alignment across modalities at an intermediate spatial scale, complementing the spot- and pathway-level objectives. The final training objective is optimized for fine-grained alignment, biological prior integration, and spatial coherence.

$$\mathcal{L} = \lambda_1 \mathcal{L}_s + \lambda_2 \mathcal{L}_p + \lambda_3 \mathcal{L}_R \tag{14}$$

Here, $\lambda_1$, $\lambda_2$, and $\lambda_3$ balance the contributions of multi-level supervision, and are set to 1, 0.25, and 0.25 respectively in our default experiments.

## 3 DATA AND EXPERIMENTS

**Dataset.** We utilize the currently most comprehensive publicly available dataset, HEST-1K (Jaume et al., 2024), as the primary training source for our model. To ensure consistency across platforms, we restrict the training data to samples generated using the Visium and Visium ST technologies. For downstream evaluation, we curate four representative datasets from HEST-1K that cover diverse tissues and experimental settings: two breast cancer datasets, including Breast Cancer (He et al., 2020) and HER2 (Andersson et al., 2021), one human dorsolateral prefrontal cortex (DLPFC) dataset (Maynard et al., 2021), and one kidney dataset (Lake et al., 2023). We use the DLPFC dataset for the zero-shot spatial layer clustering task, and apply the remaining three datasets to evaluate gene expression prediction, few-shot learning, and image-to-ST sentence retrieval.

**Implementation.** For each spatial spot, we extract a $224 \times 224$ pixel image patch centered on its coordinates. We then apply our TF–IDF strategy to select the top 250 context-relevant genes to build up the image-gene pairs. The experiments were conducted on 8 NVIDIA A100-SXM4-80GB GPUs using the OpenCLIP framework (Ilharco et al., 2021). We employed ViT-B-16 (Dosovitskiy et al., 2020) as the image encoder and a transformer-based text encoder with a maximum token length of 280. Optimization was performed using the AdamW optimizer with $\beta_1 = 0.9$, $\beta_2 = 0.95$, and a weight decay of 0.1. The initial learning rate was set to $2.56 \times 10^{-5}$ and scheduled to decay to 1% of its original value following a cosine schedule with linear warmup. The model was trained for 50 epochs with a batch size of 512. The region-aware hierarchical representation module is introduced after 25 epochs, and the region identifier is updated every 10 epochs during training. Moreover, we compile all unique gene names used across the dataset to define a collected vocabulary, treating each gene as an individual token rather than using a byte-pair encoding (BPE) tokenizer.

**Downstream Tasks.** We benchmarked our model across four downstream tasks against current SOTA methods to assess its generalization, transferability, and potential for real-world clinical applications. First, we conducted zero-shot clustering and few-shot gene expression prediction to evaluate the model's ability to transfer learned representations to unseen tissues with limited or no supervision. In addition, we included two application-oriented tasks to examine the model's practical utility. The gene expression reconstruction task involves predicting transcriptomic profiles directly from histology images, while the image-to-ST sentence retrieval task aims to match tissue image regions with their corresponding gene-level descriptions. Details of task-specific implementation are provided in Section C.2 in the supplementary material.

# 4 RESULTS

## 4.1 LOW-RESOURCE SCENARIO EVALUATION

**Few-shot Gene Expression Prediction.** We first evaluated the generalization ability of pretrained models on new datasets. For comparison, we included several pretrained foundation models as backbones, encompassing both natural image and pathology-specific representations. These include DenseNet121 (Huang et al., 2017) that is originally used by ST-Net, pretrained on natural images; and three pathology-pretrained models: CONCH (Lu et al., 2024), UNI (Chen et al., 2024), Omi-CLIP (Chen et al., 2025), and UMPIRE Han et al. (2025). To ensure a fair comparison, we adopted the training paradigm of ST-Net and kept all other settings consistent.

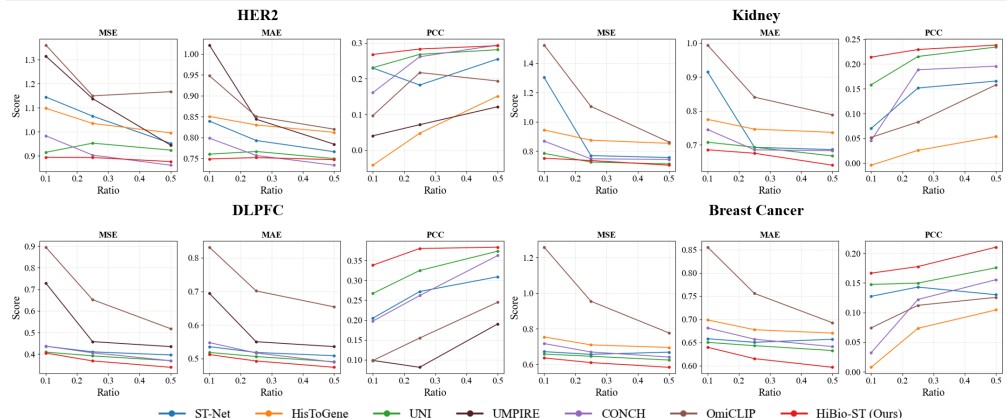

Figure 2: **Few-shot gene expression prediction.** We evaluate performance under training data ratios of 10%–50% to simulate low-resource scenarios, with each curve denoting a model.

Figure 2 illustrates model performance under varying training data ratios, highlighting the effectiveness of each approach in few-shot prediction scenarios. Each curve corresponds to a different model, with the vertical axis denoting metrics and the horizontal axis representing the ratio of training data. Across all settings, HiBio-ST consistently achieves superior generalization, with particularly notable gains at the 10% data level, where data scarcity poses significant challenges for learning reliable mappings. This strong performance in the extremely low-resource setting demonstrates HiBio-ST's remarkable transferability, enabling it to adapt effectively to previously unseen tissues with minimal supervision. In contrast, conventional models exhibit limited adaptability under such constraints, often hindered by weak inductive priors and insufficient exposure to biological heterogeneity.

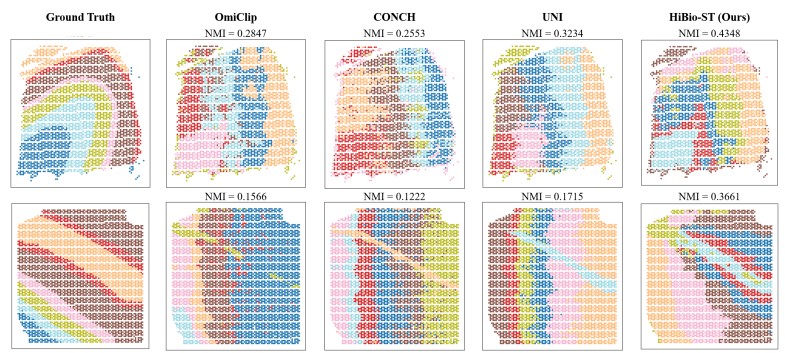

Figure 3: **Visualization of zero-shot spatial clustering results on the DLPFC dataset.** Each color denotes a distinct anatomical layer of the human dorsolateral prefrontal cortex. Higher NMI scores indicate better alignment between the predicted clusters and the ground-truth tissue layers.

**Zero-shot Layer Clustering.** The human dorsolateral prefrontal cortex (DLPFC) has a six-layered cytoarchitecture, while HER2 dataset is annotated in 4 to 7 tissue types. For fairness, we use the image encoder of each foundation model to extract patch-level embeddings, cluster them with K-Means, and apply a smoothing step to refine results (details in Supplementary C.2). Normalized mutual information (NMI) and adjusted Rand index (ARI) are then computed to quantify the alignment between predicted clusters and anatomical layers.

Table 1 summarizes the average ARI and NMI across all methods. Our HiBio-ST attains 0.264 (ARI) and 0.377 (NMI), substantially outperforming other approaches, such as STPath Huang et al. (2025) (0.177 ARI, 0.271 NMI). Figure 3 further presents the zero-shot spatial clustering results on the DLPFC dataset. HiBio-ST achieves the most accurate delineation of cortical layers, demonstrating its ability to recover fine-grained laminar structures. In contrast, competing methods fail to reliably separate adjacent layers due to the high morphological similarity and spatial continuity of DLPFC tissue sections. Figure 7 and Table 7 show that incorporating gene sentences filtered via TF-IDF improves visual features with biologically informative signals, thereby producing superior clustering performance. These results highlight the effectiveness of our multi-level alignment strategy, which leverages spatial relationships and biological priors to significantly enhance zero-shot clustering.

| Model | Vision Encoder | Parameter | DLPFC | | HER2 | |
|---|---|---|---|---|---|---|
| | | | ARI | NMI | ARI | NMI |
| Omiclip | ViT-l/16 | ∼638M | 0.1393 | 0.1947 | 0.1190 | 0.2119 |
| CONCH | ViT-b/16 | ∼200M | 0.1428 | 0.2264 | 0.1372 | 0.2288 |
| UNI | ViT-h/14 | ∼303M | 0.1647 | 0.2492 | 0.1580 | 0.2612 |
| ST-Path | GigaPath | ∼1.1B | 0.1778 | 0.2719 | 0.1544 | 0.2413 |
| UMPIRE | ViT-b/16 | ∼200M | 0.1692 | 0.2673 | 0.1452 | 0.2337 |
| HiBio-ST (Ours) | ViT-b/16 | ∼192M | 0.2640 | 0.3776 | 0.1711 | 0.2652 |

Table 1: **Zero-shot clustering performance and associated vision encoders and parameter sizes.**

## 4.2 EMPIRICAL VALIDATION ON GENE EXPRESSION PREDICTION

**Cross-Validation Evaluation.** We benchmarked our model against SOTA methods, including: (1) regression-based baselines such as ST-Net (He et al., 2020), EGN (Yang et al., 2023), HisToGene (Pang et al., 2021), His2ST (Zeng et al., 2022), TRIPLEX (Chung et al., 2024); and (2) retrieval-based approaches including BLEEP (Xie et al., 2024), mlxExpST (Min et al., 2024), OpenCLIP (Ilharco et al., 2021),and OmiCLIP (Chen et al., 2025).300 HVGs are selected for evaluation. Pearson correlation coefficient (PCC), mean squared error (MSE), and mean absolute error (MAE) are used as performance metrics. Implementation details can be found in Supplementary C.2.

| Manner | Model | HER2 | | | Breast Cancer | | | Kidney | | |
|---|---|---|---|---|---|---|---|---|---|---|
| | | MSE | MAE | PCC | MSE | MAE | PCC | MSE | MAE | PCC |
| Regression | ST-Net | 0.9225 | 0.7578 | 0.2829 | 0.6757 | 0.6656 | 0.1535 | 0.7636 | 0.6894 | 0.1551 |
| | HisToGene | 0.9596 | 0.7944 | 0.2190 | 0.6905 | 0.6685 | 0.1113 | 0.8102 | 0.7158 | 0.1114 |
| | His2ST | 1.0099 | 0.8194 | 0.0102 | 0.7084 | 0.6792 | 0.0019 | 0.8637 | 0.7437 | 0.0049 |
| | EGN | 0.9473 | 0.7883 | 0.2266 | 0.6777 | 0.6362 | 0.1270 | 0.7811 | 0.6997 | 0.1512 |
| | TRIPLEX | 0.9864 | 0.8031 | 0.0935 | 0.6834 | 0.6566 | 0.0421 | 0.7290 | 0.6757 | 0.0835 |
| Retrieval | BLEEP | 0.8920 | 0.7602 | 0.2416 | 0.7729 | 0.7124 | 0.1149 | 0.8845 | 0.7669 | 0.1432 |
| | mlxExpST | 0.9661 | 0.8705 | 0.1942 | 0.9451 | 0.7781 | 0.1013 | 1.0569 | 0.8366 | 0.1669 |
| | OpenClip | 1.0560 | 0.7862 | 0.2780 | 0.6886 | 0.6517 | 0.1333 | 0.8737 | 0.7330 | 0.1833 |
| | Omiclip | 0.8655 | 0.7986 | 0.2940 | 0.6260 | 0.6430 | 0.1930 | 0.7496 | 0.7097 | 0.2275 |
| | HiBio-ST (Ours) | 0.8298 | 0.7298 | 0.3172 | 0.6082 | 0.6074 | 0.2480 | 0.6971 | 0.6749 | 0.2627 |

Table 2: **Quantitative comparisons on gene expression prediction task.** The best performance is highlighted in orange, where we can observe that HiBio-ST outperforms the SOTAs across datasets.

The average scores of cross-validation performance for models across datasets are summarized in Table 2. Our proposed HiBio-ST consistently outperforms existing approaches almost all metrics. For example, on the Kidney dataset, characterized by high inter-sample heterogeneity, HiBio-ST achieves MSE, MAE, and PCC values of 0.6971, 0.6749, and 0.2627, respectively, representing a notable improvement over methods such as OmiCLIP (PCC = 0.2275) and HisToGene (PCC = 0.1114). These results suggest that HiBio-ST, empowered by hierarchical pretraining, is particularly

effective at capturing complex biological patterns and inter-spot relationships, which are essential for accurate modeling in heterogeneous tissue contexts.

**Pivotal Gene Expression Prediction.** The high-variance gene IGLL5 is a well-established cancer-associated biomarker. Elevated expression of IGLL5 has been significantly linked to poor patient prognosis and is believed to exert its effect by modulating tumor immune infiltration and metabolic reprogramming pathways in breast cancer (BRCA) (Feng et al., 2025; Xia et al., 2021). Figure 4 visualizes the spatial distribution of IGLL5 in WSIs and compares the predictive performance across models using PCC. Our HiBio-ST achieved the highest PCC values for IGLL5 with 0.605 and 0.456 in two representative cases, demonstrating its superior ability to reconstruct expression patterns of critical biomarkers. In contrast, other methods, such as HisToGene (-0.396) and mclSTExp (-0.371) failed to predict this key gene accurately. These results highlight the robustness of HiBio-ST in capturing expression patterns of clinically relevant markers, making it better suited for real-world deployment in pathology workflows and enhancing its translational applicability.

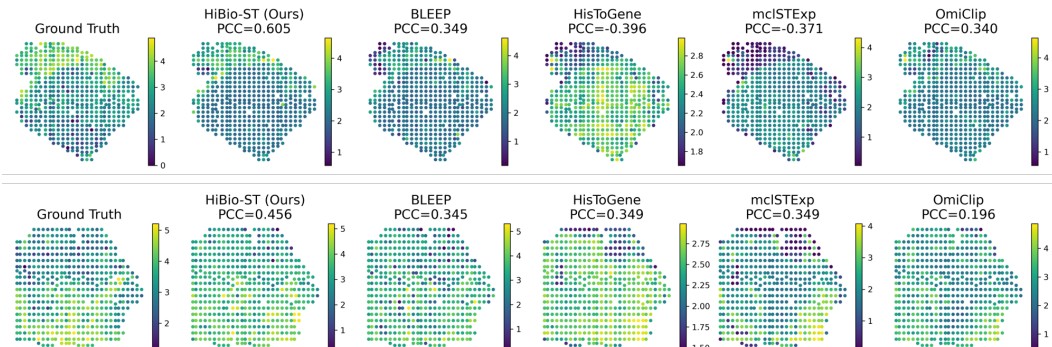

Figure 4: **Visualization of the spatial expression distribution of the cancer-associated gene IGLL5.** The ground truth and predictions are shown for representative WSIs. Our HiBio-ST model achieves the highest PCC, accurately capturing the spatial heterogeneity of IGLL5 expression.

| Model | Token Length | Breast Cancer | | | Kidney | | |
|---|---|---|---|---|---|---|---|
| | | Recall@1% | Recall@5% | Recall@10% | Recall@1% | Recall@5% | Recall@10% |
| Random | - | 0.792 | 5.032 | 8.797 | 0.928 | 4.950 | 9.171 |
| PLIP | 77 | 2.961 | 11.184 | 24.513 | 2.004 | 9.198 | 23.963 |
| OmicCLIP | 77 | 3.397 | 15.630 | 28.093 | 2.347 | 9.930 | 17.799 |
| CONCH | 128 | 2.184 | 9.669 | 17.587 | 1.469 | 7.822 | 14.919 |
| OpenCLIP | 280 | 1.303 | 8.129 | 16.563 | 1.399 | 6.600 | 12.459 |
| HiBio-ST (Ours) | 280 | 3.597 | 19.882 | 32.432 | 2.118 | 10.280 | 19.352 |

Table 3: **Image-to-sentence retrieval performance on the Breast Cancer and Kidney datasets.**

### 4.3 IMAGE-TO-ST SENTENCE RETRIEVAL

Image-to-sentence retrieval requires models to retrieve the corresponding gene expression description given a histology image, thereby assessing the semantic consistency between morphological features and molecular profiles. We evaluate retrieval performance with metrics Recall@K%.

Table 3 summarizes the image-to-sentence retrieval performance of different models on the Breast Cancer and Kidney datasets. The results demonstrate that HiBio-ST significantly outperforms existing methods, highlighting its advantage in multimodal alignment and cross-modal retrieval. Notably, unlike other foundation models pretrained on medical data, such as OmiCLIP or PLIP, which are restricted to processing short inputs (e.g., 77 tokens), HiBio-ST is capable of handling substantially longer gene sentences of up to 250 tokens. This indicates that our approach is more robust to noise introduced by long text sequences and better equipped to capture and retrieve a larger set of genes.

In addition, we introduce an updated vocabulary in which each gene is treated as an independent token, rather than relying on subword-level tokenization methods such as BPE. This design ensures that gene names are preserved in their entirety, avoiding fragmentation that can drastically reduce

the number of effectively retrievable genes. For example, in models with a 77-token input limit, BPE often splits a single gene into multiple tokens, resulting in only 20–30 genes being represented in practice. By contrast, HiBio-ST can encode and retrieve a much broader gene set, including those with relatively low expression levels but important biological significance, thereby enhancing its utility in downstream applications.

## 4.4 ABLATION STUDY

We designed two sets of ablation experiments to verify the effectiveness and necessity of each module within the HiBio-ST framework. These experiments include functional blocks in our framework, token length analysis, and hyperparameter sensitivity. We adhered to the cross-validation procedure employed in the multiple downstream tasks to ensure a fair evaluation of the ablation study.

| Functional Blocks | HER2 | | | Breast Cancer | | | Kidney | | |
|---|---|---|---|---|---|---|---|---|---|
| | MSE | MAE | PCC | MSE | MAE | PCC | MSE | MAE | PCC |
| w.o TF-IDF | 0.9025 | 0.8011 | 0.2600 | 0.7833 | 0.7307 | 0.1481 | 0.8187 | 0.7385 | 0.2087 |
| w.o Pathway | 0.8531 | 0.7626 | 0.2770 | 0.7031 | 0.7034 | 0.1689 | 0.7926 | 0.7112 | 0.2500 |
| w.o Regional Alignment | 0.8151 | 0.7407 | 0.3035 | 0.6501 | 0.6794 | 0.2162 | 0.7479 | 0.7006 | 0.2419 |
| HiBio-ST (Ours) | 0.8298 | 0.7298 | 0.3172 | 0.6082 | 0.6074 | 0.2480 | 0.6971 | 0.6749 | 0.2627 |
| Token Length=100 | 0.8250 | 0.7351 | 0.3060 | 0.6535 | 0.6968 | 0.2234 | 0.7290 | 0.6920 | 0.2620 |
| Token Length=250 | 0.8298 | 0.7298 | 0.3172 | 0.6082 | 0.6074 | 0.2480 | 0.6971 | 0.6749 | 0.2627 |
| Token Length=500 | 0.8084 | 0.7374 | 0.3379 | 0.6406 | 0.6919 | 0.2056 | 0.6816 | 0.6694 | 0.2730 |
| Pathway Ratio $\tau$=0.1 | 0.8803 | 0.7647 | 0.2776 | 0.6494 | 0.6705 | 0.2396 | 0.7241 | 0.6951 | 0.2428 |
| Pathway Ratio $\tau$=0.2 | 0.8298 | 0.7298 | 0.3172 | 0.6082 | 0.6074 | 0.2480 | 0.6971 | 0.6749 | 0.2627 |
| Pathway Ratio $\tau$=0.5 | 0.8496 | 0.7532 | 0.3032 | 0.5475 | 0.5712 | 0.2589 | 0.7178 | 0.6637 | 0.2384 |
| Loss: $\lambda_2$=0.25, $\lambda_3$=0.1 | 0.8897 | 0.7673 | 0.2854 | 0.6359 | 0.6662 | 0.2615 | 0.7210 | 0.6653 | 0.2358 |
| Loss: $\lambda_2$=0.25, $\lambda_3$=0.25 | 0.8298 | 0.7298 | 0.3172 | 0.6082 | 0.6074 | 0.2480 | 0.6971 | 0.6749 | 0.2627 |
| Loss: $\lambda_2$=0.5, $\lambda_3$=0.5 | 0.8517 | 0.7521 | 0.2781 | 0.5895 | 0.6067 | 0.2636 | 0.7193 | 0.6636 | 0.2410 |
| HVG Selection | 0.9563 | 0.8261 | 0.2580 | 0.7059 | 0.6817 | 0.1940 | 0.8344 | 0.7012 | 0.1877 |

Table 4: **Ablation study and token length analysis on gene expression prediction task.** The best performance is highlighted in orange, showing that each functional block makes a complementary contribution to the overall performance of HiBio-ST.

**Biological Information Prior.** We conduct a comprehensive ablation study to assess the contribution of each functional block within HiBio-ST, including TF-IDF–based gene sentence construction, pathway-guided supervision, and region-level alignment. We fix the token length to 250 across all ablation settings. As shown in Table 4 and Table 5, each component contributes significantly to the model's overall performance. On the HER2 dataset, for example, removing TF-IDF leads to a drop in performance from 0.7398 to 0.8425 in MSE, and from 0.3472 to 0.2700 in PCC, while the removal of pathway or regional alignment blocks results in similarly degraded metrics. This trend is consistent across the Breast Cancer and Kidney datasets, confirming the effectiveness of each module. In addition to gene expression prediction, improvements are also observed in the image-to-sentence retrieval task. As shown in Table 5, removing any individual block leads to a decline in Recall@k%, indicating that all components are essential for establishing robust cross-modal alignment. For instance, on the Breast Cancer dataset, the Recall@5% drops from 19.88 to 12.72 without TF-IDF and to 17.15 without regional alignment.

**Hierarchical Regional-wise Alignment.** Our hierarchical contrastive alignment strategy enhances the model's ability to capture spatially coherent patterns and region-level dependencies, which are critical for downstream biological interpretation. Additional Tables 7 and Figures 7 in the Supplementaty D further validate the superiority of these components in the zero-shot spatial clustering task, where HiBio-ST demonstrates a strong capacity to recover anatomically meaningful tissue layers without supervision. These results highlight the complementary role of each biological prior in bridging image and transcriptomic modalities, and validate the design of HiBio-ST in leveraging hierarchical contrastive alignment for robust and interpretable spatial omics modeling.

**Pathway Overlap Ratio.** We evaluate two configurations ($\tau = 0.1$ and $\tau = 0.5$) in comparison with our default setting ($\tau = 0.2$) to better understand the influence of hyperparameters in our framework with results summarized in Tables 4, 5, and 7. The results indicate that a low overlap

ratio leads to noticeable performance degradation. This occurs because reducing $\tau$ introduces an excessive number of positive pathway pairs, thereby increasing the probability of incorporating noisy or biologically irrelevant pathways as anchors. In contrast, when $\tau$ becomes overly strict, the performance either plateaus or slightly decreases, as the number of valid pathway anchors becomes too limited to provide sufficient biological supervision during contrastive alignment. Thus, a moderate overlap ratio yields the best trade-off between eliminating noisy anchors and retaining informative and biologically meaningful pathway anchors.

**Loss Weights Analysis.** We evaluate different configurations of the loss weights $\lambda_2$ and $\lambda_3$ in our hybrid optimization objective, with results summarized in Tables 4, 5, and 7. We observe that increasing both loss weights consistently improves performance across multiple downstream tasks, indicating that stronger supervision from biological pathway anchors provides more informative and structured guidance to the representation space. In other words, enhancing the contribution of biological priors encourages the model to more effectively align visual features with functional semantics, ultimately strengthening biologically meaningful learning. However, we also find that excessively large loss weights introduce instability during training, where gradients fluctuate and convergence becomes less reliable. This suggests that while biological priors are beneficial, overly dominating them may interfere with the balance among different learning objectives.

| Functional Blocks | Breast Cancer | | | Kidney | | |
|---|---|---|---|---|---|---|
| | Recall@1% | Recall@5% | Recall@10% | Recall@1% | Recall@5% | Recall@10% |
| w.o TF-IDF | 2.407 | 12.725 | 24.703 | 1.672 | 7.580 | 15.096 |
| w.o Pathway | 2.180 | 12.591 | 26.785 | 1.820 | 8.911 | 17.523 |
| w.o Regional Alignment | 2.834 | 17.156 | 30.667 | 1.861 | 9.276 | 18.130 |
| HiBio-ST (Ours) | 3.597 | 19.882 | 32.432 | 2.118 | 10.280 | 19.352 |
| Pathway Ratio $\tau$=0.1 | 3.407 | 18.478 | 29.720 | 1.484 | 7.535 | 16.882 |
| Pathway Ratio $\tau$=0.2 | 3.597 | 19.882 | 32.432 | 2.118 | 10.280 | 19.352 |
| Pathway Ratio $\tau$=0.5 | 4.538 | 22.322 | 32.049 | 2.009 | 10.045 | 20.149 |
| Loss: $\lambda_2$=0.25, $\lambda_3$=0.1 | 3.573 | 18.554 | 28.583 | 1.883 | 9.528 | 17.250 |
| Loss: $\lambda_2$=0.25, $\lambda_3$=0.25 | 3.597 | 19.882 | 32.432 | 2.118 | 10.280 | 19.352 |
| HVG Selection | 1.920 | 8.675 | 25.674 | 1.395 | 7.796 | 15.254 |

Table 5: **Ablation study on image-to-sentence retrieval task.**

**Token Length of Gene Sentence.** We further investigate the impact of gene sentence length on model performance by varying the token length from 100 to 500. As shown in Table 4, using a longer gene sentence (250 tokens) consistently leads to better performance across all datasets and metrics, suggesting that incorporating more gene-level information helps the model learn more comprehensive representations for downstream prediction tasks. However, increasing the token length beyond 250 introduces slight performance degradation in some settings (e.g., Breast Cancer MSE increases from 0.6082 to 0.6406), likely due to informational noise or redundancy introduced by less informative or lowly expressed genes. These results suggest a trade-off: while longer sentences enrich the biological context, excessively long sequences may overwhelm the model with irrelevant information or exceed its capacity to focus on key genes. Thus, selecting an optimal token length is crucial for balancing expressiveness and robustness in multimodal learning.

## 5 CONCLUSION

We propose a novel pretraining framework that harmonizes histological context with molecular identities in ST through progressive multi-level alignments. Moving beyond generic expression profiles, HiBio-ST distills "keyword" genes through TF–IDF weighting, while curated pathway sets offer biologically grounded anchors that inject global semantics. A spatial-wise hierarchical alignment further consolidates isolated spots into coherent regions with functional interpretability. To this end, our HiBio-ST integrates joint local vision-omics signals with functional priors and meso-scale structures, yielding a cohesive multimodal representation across sources and scales. Extensive experimental results demonstrate HiBio-ST's strong transferability and robust generalization across datasets, consistently achieving superior performance on multiple downstream tasks. Positioned as a foundation model, HiBio-ST offers a promising pathway toward transferable and scalable representations that empower real-world clinical applications and spatial omics modeling.

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

# A  DATASET DETAILS

## A.1  DETAILS IN PUBLIC DATASETS

We adopted the most comprehensive public spatial transcriptomics dataset, HEST-1K (Jaume et al., 2024), as the training set. To ensure cross-platform consistency and mitigate noise issues often present in high-resolution datasets, we restricted the training samples to those generated using Visium and Visium ST technologies, resulting in a total of 949 slides and approximately 1.5 million spots collected from 149 cohorts, covering 26 organs across two species (Homo sapiens and Mus musculus). The spot size in these datasets ranges from 55 $\mu$m to 100 $\mu$m.

For downstream evaluations, we employed the HER2 (Andersson et al., 2021), Breast Cancer (He et al., 2020), and Kidney (Lake et al., 2023) datasets for the few-shot prediction, image-to-ST sentence retrieval, and gene expression prediction tasks, while the DLPFC dataset (Maynard et al., 2021) was used for the zero-shot cortical layer clustering task. Specifically, the HER2 dataset contains 8 samples with 36 WSIs and a total of 13,620 spots (176–712 spots per slide), the Breast Cancer dataset includes 22 samples with 66 WSIs and 30,066 spots (256–712 spots per slide), and the Kidney dataset consists of 22 samples with 23 WSIs and 25,944 spots (317–4,166 spots per slide). For the DLPFC dataset, we selected 2 samples with 8 slides (IDs: 151507–151510 and 151673–151676) for testing. The spot size of HER2 and Breast Cancer datasets is 100 $\mu$m,

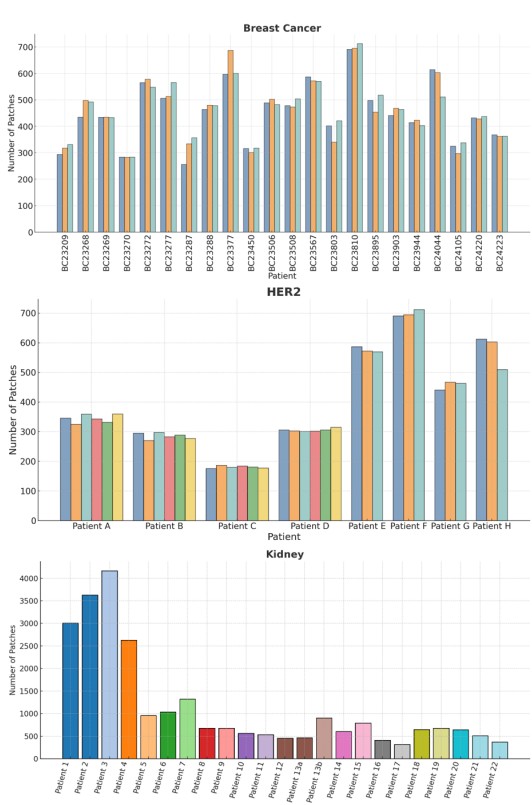

Figure 5: Profiles of public datasets.

whereas that of the DLPFC and Kidney datasets is 55 $\mu$m. Figure 5 provides detailed information about each sample used in the three datasets, along with their summarized profiles.

## A.2  GENE SELECTION AND PREPROCESSING.

**Gene Selection.** Following the methodology outlined in BLEEP (Xie et al., 2024), we selected the top 300 genes with the highest average variance expression across datasets for prediction. This ensures that the selected set captures informative variation while filtering out low-variance genes that contribute little to generalization. For preprocessing, we directly applied a $\log(1+x)$ transformation to the raw count matrices before training, which helps stabilize variance and alleviate the skewness commonly observed in count data. Figure 6 provides an overview of the selected genes and the corresponding dataset codes.

**Vocabulary Unification.** Due to the inclusion of samples from both *Homo sapiens* and *Mus musculus*, as well as cohort-specific technical and platform differences that result in variation in both the number and identity of measured genes, unified modeling across the dataset presents a significant challenge. To mitigate this, we construct a shared human gene vocabulary that serves as the common target space for aligning all expression matrices. For mouse-derived samples, orthologous gene mappings are retrieved from the Ensembl database (Cunningham et al., 2022) using the biomaRt interface (Durinck et al., 2009), thereby enabling each mouse gene to be mapped to its human counterpart. Genes without valid orthologs are discarded to avoid introducing noise, while missing human genes in a given sample are imputed with zeros to maintain consistency in dimensionality. After alignment, all mouse and human matrices are vertically concatenated, producing a

unified gene-by-spot expression matrix. This harmonized representation ensures that cross-species and cross-cohort samples are directly comparable, facilitating large-scale model training across heterogeneous datasets.

**Pre-processing Procedure.** For each spatial spot, we first crop a $224 \times 224$ pixel histological image patch centered on its spatial coordinates to preserve local morphological context. In parallel, we construct a unified gene vocabulary by merging annotations from all cohorts and harmonizing gene identifiers across species and platforms. To capture biologically meaningful information, we apply a TF–IDF strategy, selecting the top 250 most context-relevant genes per spot. The selected gene set is then organized into a textual sentence, which is further enriched with spot-level metadata such as organ type and disease status, enabling the model to incorporate tissue- and condition-specific context better. To ensure consistency in representation, we compile all unique gene names encountered across the dataset to form a fixed vocabulary, treating each gene as an atomic token. Unlike natural language processing pipelines that employ subword tokenizers (e.g., byte-pair encoding), our design explicitly treats each gene as a distinct unit, thereby preserving biological semantics without fragmenting gene symbols into arbitrary substrings.

| Dataset | Genes to be predicted |
|---|---|
| **Breast Cancer** | 'TFF3', 'IGLL5', 'ERBB2', 'MUC1', 'MGP', 'RPS3', 'S100A9', 'KRT19', 'H2AJ', 'NHERF1', 'COX6C', 'AZGP1', 'TMSB10', 'IFI27', 'RPL18A', 'B2M', 'C4B', 'FASN', 'RPL38', 'RPL19', 'APOE', 'HLA-B', 'HSPB1', 'ACTB', 'PABPC1', 'RPS15', 'RPS12', 'FN1', 'IGFBP2', 'RPS8', 'HLA-C', 'FTH1', 'CCND1', 'S100A8', 'CTSD', 'RPL31', 'RPL37A', 'EEF2', 'KRT18', 'XBP1', 'TAGLN', 'RPS17', 'RPS21', 'COL1A1', 'BEST1', 'SNHG25', 'ACTG1', 'RPL18', 'RPS18', 'RPS2', 'CST3', 'HLA-A', 'PSMD3', 'UBA52', 'SSR4', 'RPL29', 'RPL13', 'A2M', 'RPS6', 'AEBP1', 'GAPDH', 'RPS4X', 'RPL30', 'FTL', 'RPS11', 'RPLP1', 'RPL8', 'RPS24', 'RPS28', 'RPL28', 'RPS27A', 'RPL10', 'RPS19', 'BST2', 'RPL37', 'RPS27', 'S100A11', 'CD74', 'CLDN3', 'EVL', 'KRT8', 'RPL7A', 'RPS14', 'RPL35', 'PPP1CA', 'P4HB', 'SYNGR2', 'HSP90AB1', 'HLA-DRA', 'SLC39A6', 'RPLP0', 'RPL27A', 'RPL23', 'RPS29', 'SERF2', 'RPL12', 'LY6E', 'RPS5', 'RACK1', 'GNAS', 'LGALS3BP', 'S100A14', 'MZT2B', 'PFN1', 'IFITM3', 'RPL11', 'HLA-E', 'RPL35A', 'H3-3B', 'RPL9', 'RPL13A', 'MIEN1', 'PRDX2', 'GPX4', 'RPL15', 'APOC1', 'BGN', 'CLU', 'PSAP', 'ATP6V0B', 'C1QA', 'ISG15', 'TPT1', 'RPL36', 'IFI6', 'RPL23A', 'RPL27', 'RPS9', 'FXYD3', 'C3', 'GATA3', 'KRT7', 'BSG', 'MZT2A', 'PTMA', 'PLXNB2', 'EIF4A1', 'STARD10', 'IGFBP5', 'RPL3', 'TAPBP', 'RPS16', 'SPDEF', 'NDUFA13', 'ELF3', 'RPS3A', 'ALDOA', 'TAGLN2', 'CD81', 'TMSB4X', 'ATP6AP1', 'COL1A2', 'BCAP31', 'MALAT1', 'DDX5', 'TUBA1B', 'CFL1', 'MYL9', 'ATP1A1', 'PTPRF', 'RPS23', 'RPL5', 'ANPEP', 'CALR', 'RPL32', 'NBEAL1', 'FAU', 'GNB2', 'PRSS8', 'RPLP2', 'S100A6', 'RPS15A', 'SDC1', 'RPS7', 'UQCRQ', 'SLC25A6', 'PRDX1', 'VIM', 'RPL14', 'GAS5', 'RPL4', 'TUFM', 'POLR2L', 'NR2F6', 'RPL34', 'TMEM123', 'GRINA', 'UBC', 'H1-10', 'IER3', 'OAZ1', 'METRN', 'IGFBP4', 'RPS15A', 'ENO1', 'CISD3', 'PSMB3', 'POSTN', 'RPL7', 'TIMP1', 'FLNA', 'CD24', 'RPS25', 'CRIP2', 'LAPTM5', 'NDUFB9', 'ARHGDIA', 'JUP', 'GSTP1', 'CLDN4', 'COL3A1', 'CYBA', 'EDF1', 'RPL24', 'EIF3C', 'NDUFA11', 'COX8A', 'NPDC1', 'SPARC', 'RPS20', 'CD63', 'MFGE8', 'GLUL', 'PPDPF', 'SREBF1', 'MAGED2', 'DEGS2', 'C12orf57', 'RPSA', 'SPINT2', 'RPL41', 'RHOC', 'SLC44A2', 'SERINC2', 'PTMS', 'COX6B1', 'CTSB', 'UBB', 'WDR83OS', 'HNRNPA2B1', 'ATP5F1E', 'C1QB', 'GRN', 'COL6A2', 'SEC61A1', 'ATG10', 'EIF5A', 'CRABP2', 'ELOB', 'TPI1', 'COX4I1', 'SELENOW', 'TXNIP', 'UQCR11', 'RPL6', 'MMP14', 'MDK', 'GUK1', 'TUBB', 'H1-2', 'LMNA', 'RPL22', 'MYL6', 'CHCHD2', 'CTTN', 'RPL10A', 'EEF1A1', 'LSM4', 'DHCR24', 'CAPNS1', 'LAPTM4A', 'SSR2', 'UBE2M', 'TSPO', 'LMAN2', 'RPS10', 'PFDN5', 'NENF', 'TSKU', 'SERPINH1', 'IGLL1', 'LGALS1', 'CHPF', 'EEF1D', 'MIDN', 'ATP5MC2', 'CLDN7', 'STARD3', 'LLGL2', 'CCT3', 'EIF3B', 'ZNF90', 'SH3BGRL3', 'MZB1', 'CDC37', 'PTBP1', 'AP2M1', 'IGFBP7', 'ENSA', 'EIF4G1' |
| **HER2** | 'IGKC', 'IGHG3', 'IGLC2', 'MUCL1', 'IGHM', 'IGHA1', 'ERBB2', 'IGLC3', 'CALML5', 'IGHG4', 'PTPRF', 'FN1', 'ACTG1', 'PSMB3', 'MIEN1', 'TPT1', 'S100A9', 'CISD3', 'BEST1', 'COL1A2', 'IGHG1', 'FASN', 'COL1A1', 'ACTB', 'B2M', 'TFF3', 'GRB7', 'APOE', 'FTH1', 'PCGF2', 'MGP', 'PABPC1', 'SERF2', 'PSMD3', 'COL3A1', 'S100A8', 'DDX5', 'NDUFB9', 'HLA-B', 'S100A6', 'GNAS', 'TMSB10', 'EEF2', 'ADAM15', 'MLLT6', 'KRT81', 'AZGP1', 'TRAF4', 'TUBA1B', 'CD24', 'LASP1', 'S100A14', 'SCD', 'KRT7', 'STARD3', 'TAGLN2', 'PERP', 'AEBP1', 'CD74', 'PTMS', 'HLA-DRA', 'CRIP2', 'IFI27', 'PPP1R1B', 'VMP1', 'PFN1', 'FTL', 'COX7C', 'GNB2', 'EEF1D', 'MYL6', 'P4HB', 'FNBP1L', 'GAPDH', 'FAU', 'SPINT2', 'PRDX1', 'ATP1A1', 'SYNGR2', 'XBP1', 'CCT3', 'MMP14', 'HNRNPA2B1', 'COX6C', 'ORMDL3', 'EIF4G1', 'CALR', 'PSAP', 'KRT18', 'PLXNB2', 'CTSD', 'SPARC', 'SSR4', 'HLA-C', 'CLDN3', 'GRINA', 'KRT19', 'APOC1', 'SLC25A6', 'HLA-A', 'EIF4G2', 'RACK1', 'HSP90AB1', 'TAGLN', 'FLNA', 'VIM', 'C1QA', 'ATG10', 'HSP90AA1', 'S100A11', 'BGN', 'HSPB1', 'HLA-E', 'C3', 'CNN3', 'ATP6V0B', 'PRRC2A', 'SPDEF', 'UQCRQ', 'MDK', 'COMP', 'SDC1', 'LMNA', 'POSTN', 'CD63', 'JTB', 'CHCHD2', 'HSPA8', 'IGFBP2', 'OAZ1', 'TSPO', 'IDH2', 'MUC1', 'LLGL2', 'PGAP3', 'CD99', 'PRSS8', 'MIDN', 'CIB1', 'TUBB', 'A2M', 'ALDOA', 'SCAND1', 'COPS9', 'MZT2B', 'LUM', 'LY6E', 'LGALS3', 'CLDN4', 'ELOVL1', 'CST3', 'CFL1', 'CALM2', 'NACA', 'COL6A2', 'PSMB4', 'DBI', 'LAPTM4A', 'GPX4', 'PIP4K2B', 'ERGIC1', 'SUPT6H', 'NUCKS1', 'SF3B5', 'DHCR24', 'TXNIP', 'MYL9', 'PEBP1', 'LAPTM5', 'UBA52', 'HSP90B1', 'SEC61A1', 'MMACHC', 'MAP3K12', 'MAPKAPK2', 'PNMT', 'CTSB', 'NBL1', 'DDIT4', 'FUCA2', 'JUP', 'PLD3', 'BSG', 'NUPR1', 'LGALS1', 'LGALS3BP', 'UBC', 'SNRPB', 'PTMA', 'BST2', 'PCSK7', 'ISG15', 'KDELR1', 'IGFBP7', 'COX6B1', 'UBE2M', 'COX4I1', 'PPP1CA', 'COL18A1', 'COPE', 'CYBA', 'PTGES3', 'MYH9', 'TAPBP', 'CD81', 'EDF1', 'SSR2', 'CLTC', 'HEBP2', 'RALY', 'PPDPF', 'ZYX', 'HINT1', 'INTS1', 'PCBP2', 'TMBIM6', 'ZBTB7B', 'COX5B', 'ACTN4', 'ZFP36L1', 'FADS2', 'CTTN', 'TIMP1', 'LMAN2', 'STARD10', 'VCP', 'CHPF', 'C12ORF57', 'UBL5', 'GUK1', 'DCN', 'PFDN5', 'SLC2A4RG', 'C1QB', 'S100A10', 'ENO1', 'TMSB4X', 'KRT8', 'SREBF1', 'CLDN7', 'PKM', 'SRRM2', 'ARHGDIA', 'IFI6', 'NDUFB7', 'ABCD3', 'PPP1R14B', 'ROMO1', 'FKBP2', 'PFKL', 'KDELR2', 'NDUFA3', 'MRPL12', 'LSM7', 'C1ORF122', 'CPD', 'SMARCD2', 'NDUFA4', 'LDHB', 'RABAC1', 'ATP6AP1', 'ITM2B', 'SEC61G', 'SERINC2', 'RRBP1', 'EPCAM', 'PGK1', 'HLA-DPA1', 'HM13', 'TMED9', 'UQCR11', 'CYB561', 'TSTD1', 'PTBP1', 'LPCAT1', 'TRIM28', 'UBB', 'AP2S1', 'SH3BGRL3', 'EIF3B', 'AKT1', 'PSMD8', 'SDF4', 'CCDC152', 'MCL1', 'PSMB1', 'UQCR10', 'LSM4', 'EIF4EBP1', 'CD46', 'PDIA6', 'SOD1', 'APP', 'TYMP', 'AP1S1', 'CSDE1', 'ARPC1B', 'CD9', 'POLR2L', 'CCND1' |
| **Kidney** | 'IGKC', 'UMOD', 'MT-CO1', 'GPX3', 'SPP1', 'MT-CO2', 'DEFB1', 'IGHG3', 'IGHG4', 'ALDOB', 'ATP6V0C', 'IGLC2', 'IGHG1', 'IGHA1', 'RPS23', 'MT-CO3', 'RPS12', 'MT-ND3', 'RPLP1', 'RPS27', 'RPS18', 'RPL34', 'RPS15A', 'MT-ND1', 'RPL37A', 'MT-CYB', 'SLC12A1', 'MT1G', 'MT-ATP6', 'RPS27A', 'RPL37', 'RPL41', 'IGFBP7', 'APOE', 'RPS15', 'RPL12', 'RPL17', 'EEF1A1', 'RPL26', 'RPS21', 'RPS24', 'ATP1A1', 'RPS28', 'TMSB4X', 'WFDC2', 'RPL32', 'RPL30', 'PCK1', 'RPS4X', 'RPS8', 'RPS14', 'RPL11', 'RPS13', 'C7', 'MT-ND2', 'RPL19', 'RPS7', 'A2M', 'RPS29', 'RPS25', 'MGP', 'RPS16', 'RPS6', 'RPS2', 'MT-ND4', 'RPL9', 'ATP1B1', 'RPL7A', 'FAU', 'RPL35A', 'IGLC3', 'DZK1IP1', 'RPL28', 'TPT1', 'RPL10', 'PIGR', 'RPLP2', 'RPL15', 'MT-ND5', 'FTH1', 'COX7C', 'RPS3A', 'RPL23A', 'HLA-C', 'RPS19', 'HSD11B2', 'TMSB10', 'GATM', 'RPL5', 'RPL38', 'ATP5F1E', 'RPS26', 'RPL31', 'SERF2', 'UQCRQ', 'KNG1', 'AQP2', 'SOD2', 'TAGLN', 'RPL24', 'RPL35', 'RPL13A', 'PTH1R', 'RPL21', 'PPIA', 'S100A6', 'RPS20', 'RPS9', 'RPL3', 'RPLP0', 'RPL18', 'RPS10', 'AQP1', 'RPL36A', 'NDUFA4', 'COX4I1', 'SPARC', 'HLA-DRA', 'CST3', 'VIM', 'UQCR11', 'RPS3', 'RPL10A', 'DCN', 'CD24', 'COX5B', 'CD74', 'RPL29', 'RPL6', 'SERPINA1', 'RPS17', 'CKB', 'GAPDH', 'SLC12A3', 'COX6A1', 'IGLC1', 'FTL', 'B2M', 'RPL8', 'COX6B1', 'MALAT1', 'CTSB', 'RPL18A', 'RPL22', 'GSTP1', 'COX8A', 'RPS11', 'ATP5MG', 'UQCRB', 'HLA-A', 'CHCHD10', 'RPL7', 'HLA-B', 'ITM2B', 'ACTG1', 'IGFBP5', 'LDHB', 'COL1A2', 'CDH16', 'RPL23', 'FXYD2', 'PODXL', 'RPL13', 'RPL14', 'CD81', 'NDUFB2', 'S100A11', 'CALB1', 'IFITM3', 'SELENOP', 'EEF1G', 'RPL27', 'RPL36', 'COX6C', 'ATP5MC3', 'ADIRF', 'MUC1', 'PTGDS', 'GNAS', 'CLCNKB', 'COX7B', 'CTSD', 'MYL9', 'MT2A', 'MAL', 'RACK1', 'ACTB', 'MT-ND4L', 'CA12', 'NAT8', 'CRYAB', 'ADGRG1', 'COX7A2', 'PEBP1', 'EIF1', 'PSAP', 'UBA52', 'REN', 'ATP5F1B', 'NDUFA1', 'ATP5ME', 'RPSA', 'TIMP3', 'NEAT1', 'ATP5F1A', 'UBL5', 'MT1E', 'NACA', 'MMP7', 'COL18A1', 'CD63', 'TIMP1', 'HINT1', 'ENO1', 'ELOB', 'UQCRH', 'UBB', 'RPL27A', 'RPS5', 'MIF', 'MYL6', 'AEBP1', 'TOMM7', 'ATP5PF', 'ACTA2', 'SLC25A6', 'CLU', 'SLC25A5', 'SRP14', 'GPX4', 'S100A10', 'MIOX', 'BSG', 'NDUFB1', 'CTSH', 'PFDN5', 'SLC3A1', 'S100A2', 'PKM', 'TSPAN1', 'LUM', 'RNASE1', 'ASS1', 'APP', 'BGN', 'HSPB1', 'RPL4', 'COL3A1', 'TMA7', 'SLC5A12', 'SERPINA5', 'IFITM2', 'SERPING1', 'SPINK1', 'UQCR10', 'PFN1', 'CHCHD2', 'UBC', 'CXCL14', 'TMBIM6', 'SAT1', 'LRP2', 'APLP2', 'CFL1', 'SLC13A3', 'EEF2', 'CD59', 'HLA-DRB1', 'CALR', 'PPP1R1A', 'NME2', 'TPI1', 'ATP6V0B', 'GSTM3', 'COX5A', 'ANXA2', 'TXNIP', 'PRDX5', 'NDUFB7', 'ATP5F1D', 'OST4', 'PRDX1', 'ATP5MC2', 'SLC25A3', 'CYB5A', 'MYL12B', 'OAZ1', 'NDUFB9', 'ATP5PO', 'CYSTM1', 'TUBA1B', 'ANPEP', 'HSP90AA1', 'FABP1', 'PPDPF', 'IGFBP4', 'DSTN', 'EPAS1', 'RNASEK', 'PTMA' |
| **DLPFC** | 'IGKC', 'UMOD', 'MT-CO1', 'GPX3', 'SPP1', 'MT-CO2', 'DEFB1', 'IGHG3', 'IGHG4', 'ALDOB', 'ATP6V0C', 'IGLC2', 'IGHG1', 'IGHA1', 'RPS23', 'MT-CO3', 'RPS12', 'MT-ND3', 'RPLP1', 'RPS27', 'RPS18', 'RPL34', 'RPS15A', 'MT-ND1', 'RPL37A', 'MT-CYB', 'SLC12A1', 'MT1G', 'MT-ATP6', 'RPS27A', 'RPL37', 'RPL41', 'IGFBP7', 'APOE', 'RPS15', 'RPL12', 'RPL17', 'EEF1A1', 'RPL26', 'RPS21', 'RPS24', 'ATP1A1', 'RPS28', 'TMSB4X', 'WFDC2', 'RPL32', 'RPL30', 'PCK1', 'RPS4X', 'RPS8', 'RPS14', 'RPL11', 'RPS13', 'C7', 'MT-ND2', 'RPL19', 'RPS7', 'A2M', 'RPS29', 'RPS25', 'MGP', 'RPS16', 'RPS6', 'RPS2', 'MT-ND4', 'RPL9', 'ATP1B1', 'RPL7A', 'FAU', 'RPL35A', 'IGLC3', 'DZK1IP1', 'RPL28', 'TPT1', 'RPL10', 'PIGR', 'RPLP2', 'RPL15', 'MT-ND5', 'FTH1', 'COX7C', 'RPS3A', 'RPL23A', 'HLA-C', 'RPS19', 'HSD11B2', 'TMSB10', 'GATM', 'RPL5', 'RPL38', 'ATP5F1E', 'RPS26', 'RPL31', 'SERF2', 'UQCRQ', 'KNG1', 'AQP2', 'SOD2', 'TAGLN', 'RPL24', 'RPL35', 'RPL13A', 'PTH1R', 'RPL21', 'PPIA', 'S100A6', 'RPS20', 'RPS9', 'RPL3', 'RPLP0', 'RPL18', 'RPS10', 'AQP1', 'RPL36A', 'NDUFA4', 'COX4I1', 'SPARC', 'HLA-DRA', 'CST3', 'VIM', 'UQCR11', 'RPS3', 'RPL10A', 'DCN', 'CD24', 'COX5B', 'CD74', 'RPL29', 'RPL6', 'SERPINA1', 'RPS17', 'CKB', 'GAPDH', 'SLC12A3', 'COX6A1', 'IGLC1', 'FTL', 'B2M', 'RPL8', 'COX6B1', 'MALAT1', 'CTSB', 'RPL18A', 'RPL22', 'GSTP1', 'COX8A', 'RPS11', 'ATP5MG', 'UQCRB', 'HLA-A', 'CHCHD10', 'RPL7', 'HLA-B', 'ITM2B', 'ACTG1', 'IGFBP5', 'LDHB', 'COL1A2', 'CDH16', 'RPL23', 'FXYD2', 'PODXL', 'RPL13', 'RPL14', 'CD81', 'NDUFB2', 'S100A11', 'CALB1', 'IFITM3', 'SELENOP', 'EEF1G', 'RPL27', 'RPL36', 'COX6C', 'ATP5MC3', 'ADIRF', 'MUC1', 'PTGDS', 'GNAS', 'CLCNKB', 'CTSD', 'MYL9', 'MT2A', 'MAL', 'RACK1', 'ACTB', 'MT-ND4L', 'CA12', 'NAT8', 'CRYAB', 'ADGRG1', 'COX7A2', 'PEBP1', 'EIF1', 'PSAP', 'UBA52', 'REN', 'ATP5F1B', 'NDUFA1', 'ATP5ME', 'RPSA', 'TIMP3', 'NEAT1', 'ATP5F1A', 'UBL5', 'MT1E', 'NACA', 'MMP7', 'COL18A1', 'CD63', 'TIMP1', 'HINT1', 'ENO1', 'ELOB', 'UQCRH', 'UBB', 'RPL27A', 'RPS5', 'MIF', 'MYL6', 'AEBP1', 'TOMM7', 'ATP5PF', 'ACTA2', 'SLC25A6', 'CLU', 'SLC25A5', 'SRP14', 'GPX4', 'S100A10', 'MIOX', 'BSG', 'NDUFB1', 'CTSH', 'PFDN5', 'SLC3A1', 'S100A2', 'PKM', 'TSPAN1', 'LUM', 'RNASE1', 'ASS1', 'APP', 'BGN', 'HSPB1', 'RPL4', 'COL3A1', 'TMA7', 'SLC5A12', 'SERPINA5', 'IFITM2', 'SERPING1', 'SPINK1', 'UQCR10', 'PFN1', 'CHCHD2', 'UBC', 'CXCL14', 'TMBIM6', 'SAT1', 'LRP2', 'APLP2', 'CFL1', 'SLC13A3', 'EEF2', 'CD59', 'HLA-DRB1', 'CALR', 'PPP1R1A', 'NME2', 'TPI1', 'ATP6V0B', 'GSTM3', 'COX5A', 'ANXA2', 'TXNIP', 'PRDX5', 'NDUFB7', 'ATP5F1D', 'OST4', 'PRDX1', 'ATP5MC2', 'SLC25A3', 'CYB5A', 'MYL12B', 'OAZ1', 'NDUFB9', 'ATP5PO', 'CYSTM1', 'TUBA1B', 'ANPEP', 'HSP90AA1', 'FABP1', 'PPDPF', 'IGFBP4', 'DSTN', 'EPAS1', 'RNASEK', 'PTMA' |

Figure 6: **Genes selection in each public dataset.** This figure showcases the top 300 high-variance genes for each public dataset utilized in gene expression prediction and few-shot prediction task.

# B    RELATED WORK

## B.1    DEEP LEARNING APPROACHES FOR IMAGE-ST TASKS

Recent advances in artificial intelligence have provided powerful tools for medical image analysis (Chen et al., 2021; Ke et al., 2023), spurring the development of diverse deep learning strategies for integrating histology with spatial transcriptomics. Early efforts such as ST-Net (He et al., 2020) employed transfer learning with DenseNet121 (Huang et al., 2017) to build a patch-to-spot regression model for expression prediction. Subsequent work has evolved along two main directions: one leverages image similarity, as in EGN (Yang et al., 2023) which dynamically selects exemplar points, or BLEEP (Xie et al., 2024) which aligns bi-modal embeddings for profile matching; the other exploits spatial context, where HisToGene (Pang et al., 2021) and His2ST (Zeng et al., 2022) integrate Transformers and graph neural networks to capture neighborhood dependencies.

Beyond expression inference, images have also been integrated into downstream ST tasks. iIM-PACT (Jiang et al., 2024) constructs histology-based spatial domains and identifies domain-specific differentially expressed genes, while stMMC (Li et al., 2024) uses contrastive learning to jointly embed histological and transcriptomic features for spatial clustering. GIST (Ge et al., 2025) combines histology and gene expression for spatial cellular profiling, and ScribbleDom (Rahman et al., 2023) employs semi-supervised learning where scribble annotations guide spatial domain identification. Together, these approaches highlight the expanding role of deep learning in image–ST integration, ranging from gene expression prediction to domain detection, cell-type decomposition, and multi-modal representation learning.

However, despite these advances, current models remain largely task-specific and are typically trained on relatively small and homogeneous datasets, which limits their generalizability across tissues, disease types, and experimental platforms. Moreover, their design is often tied to a single downstream objective, restricting transferability to new tasks and reducing robustness when applied to unseen data. These limitations call for a more general foundation model paradigm that can capture universal histology–transcriptomics correspondences and provide transferable representations across diverse biological contexts.

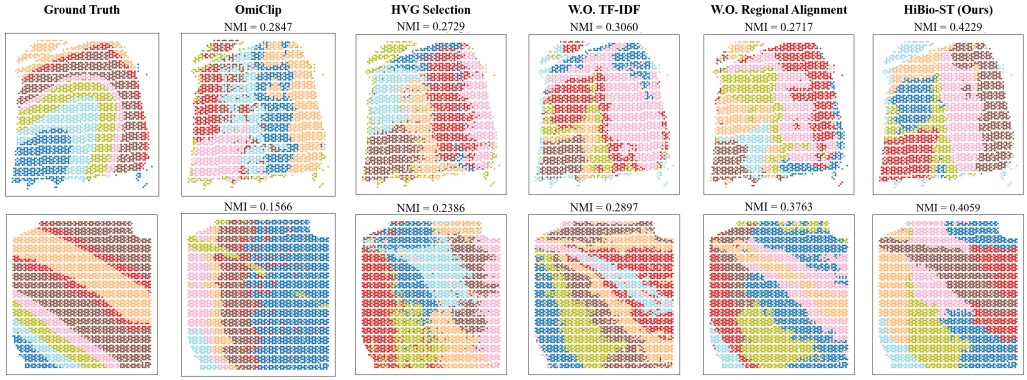

Figure 7: **Visualization of ablation study on zero-shot clustering on the DLPFC dataset.** Each color denotes an anatomical layer of the human dorsolateral prefrontal cortex. Higher NMI scores indicate better alignment between the predicted clusters and the ground-truth tissue layers.

## B.2    FOUNDATION MODELS IN PATHOLOGY AND SPATIAL TRANSCRIPTOMICS

The success of foundation models in computer vision and natural language processing has inspired analogous efforts in computational pathology and spatial transcriptomics. In digital pathology, large-scale pretrained models such as CONCH (Lu et al., 2024), PLIP (Huang et al., 2023), and UNI (Chen et al., 2024) have demonstrated that transformer-based architectures trained on millions of whole-slide images can yield transferable representations for diverse downstream tasks, including cancer subtyping, biomarker discovery, and survival prediction. These advances highlight the potential of large-scale pretraining to overcome task-specific limitations and data scarcity in pathology.

In the molecular domain, SpatialAgent Wang et al. (2025b) targets broader spatial analysis and decision-making workflows, particularly task-driven gene panel design. It emphasizes selecting genes based on task-specific utility rather than building a transferable pretraining vocabulary. GenePT Chen & Zou (2024), by contrast, leverages public gene descriptions and large language models to derive semantic embeddings for individual genes, constructing cell-level representations by ranking genes according to expression levels.

Recent attempts in the spatial transcriptomics domain have introduced foundation model settings that leverage multimodal pretraining. OmiCLIP (Chen et al., 2025) employs contrastive learning to align histology patches with transcriptomic profiles, providing a first step toward vision–omics integration. Meanwhile, scGPT-spatial (Wang et al., 2025a) adapts large language models to spatial omics by continual pretraining on millions of spots, but it operates purely on transcriptomic input without leveraging histological context.

However, these works underscore both the promise and the current limitations of foundation models for ST. While they demonstrate the feasibility of large-scale multimodal pretraining, existing approaches either over-rely on top-expressed genes with limited biological specificity or treat each spot as an isolated unit, overlooking spatial dependencies. This motivates the development of new paradigms that combine biologically meaningful priors with hierarchical modeling to achieve more robust and generalizable representations.

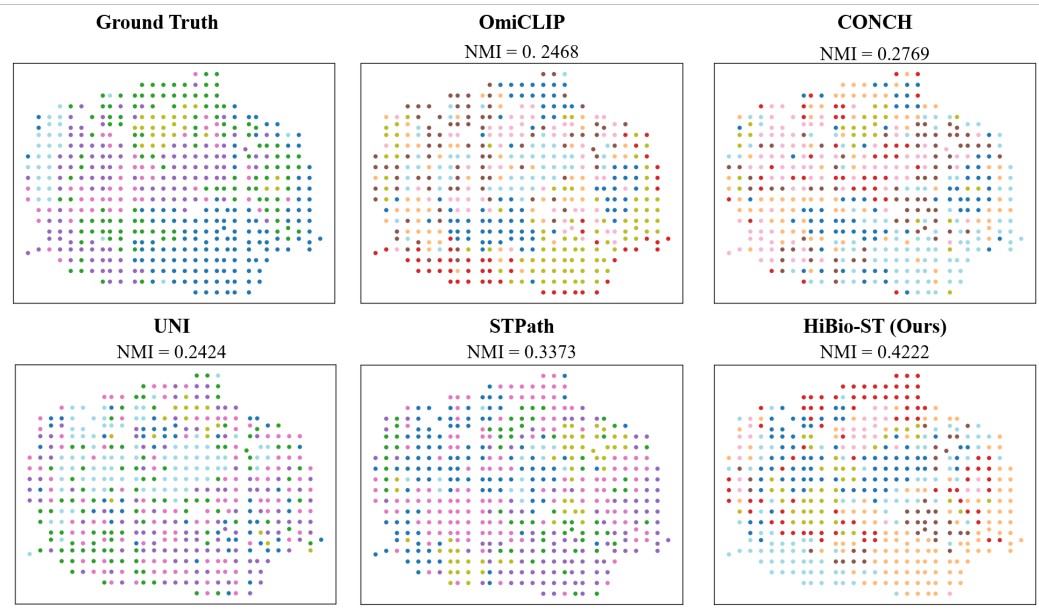

Figure 8: **Visualization of zero-shot spatial clustering results on HER2 dataset.**

## C    Implementation Details

### C.1    Data Augmentation

To ensure consistency across both training and downstream tasks, we adopted a unified data augmentation strategy designed to improve model generalization. Specifically, we applied random horizontal and vertical flips to account for variations in image orientation, random 90-degree rotations to enhance rotational invariance, and a final normalization step to standardize pixel values according to the distribution used by pretrained models (mean [0.485, 0.456, 0.406] and standard deviation [0.229, 0.224, 0.225]).

## C.2    Implementation Details for Downstream Tasks

Our work aims to leverage easily accessible histology images to analyze spatial transcriptomics, thereby reducing the dependency on costly ST profiling. To comprehensively evaluate the biological utility of our histology-derived representations, we design a series of downstream benchmarking tasks that reflect distinct biological objectives. Below, we clarify the specific purpose and biological interpretation of each task.

**(1) Zero-shot Spatial Clustering.**

*Purpose:* This task demonstrates the model's ability to produce biologically meaningful spatial structures directly from histology-derived embeddings without any supervision Maynard et al. (2021). It examines whether the pretrained model can implicitly generate and organize spatial representations that correspond to real tissue architectures.

*Biological Interpretation:* A model capable of zero-shot clustering that aligns with known tissue architectures indicates that its histology-based representations preserve biologically meaningful spatial patterns, reflecting the ability to capture inherent tissue heterogeneity purely from visual cues.

**(2) Few-shot Prediction & Gene Expression Prediction.**

*Purpose:* Few-shot prediction assesses the generalization ability of a foundation model under limited ST supervision or with only minimal annotated data. Gene expression prediction further evaluates the model's ability to infer ST profiles directly from histology images, examining how effectively visual morphology can recover underlying molecular signals He et al. (2020).

*Biological Interpretation:* These tasks reflect practical scenarios where only a small portion of the tissue is profiled with ST. This demonstrates how histology can serve as a surrogate to recover biologically meaningful molecular signals when ST data is sparse or unavailable.

**(3) Image-to-ST Sentence Retrieval.**

*Purpose:* This task evaluates the model's ability to align and maintain consistency between tissue morphology and higher-level biological functions Chen et al. (2025).

*Biological Interpretation:* Each gene set represents a distinct biological process or phenotype. Successful retrieval indicates that the model effectively learns biologically interpretable visual–molecular relationships and can reliably associate histological regions with their relevant functional gene sets.

### C.2.1    Zero-shot Layer Clustering Task.

For each histological slide, we perform unsupervised clustering solely based on image features. This process does not rely on any textual input or downstream annotations, but instead uses patch-level representations extracted by the trained image encoder for spatial clustering and post-processing.

To ensure a fair comparison across different foundation models, for each spatial spot, we first extract a visual representation vector from its corresponding image patch using the image encoder, denoted as $\mathbf{z}_i \in \mathbb{R}^d$. To mitigate scale differences and enhance clustering robustness, each vector is normalized with the $\ell_2$ norm to obtain unit features.

| Model | ARI | NMI |
|---|---|---|
| Omiclip | 0.1220 | 0.1747 |
| CONCH | 0.1346 | 0.2070 |
| UNI | 0.1322 | 0.2127 |
| ST-Path | 0.1695 | 0.2568 |
| HiBio-ST (Ours) | 0.2503 | 0.3542 |

Table 6: Zero-shot clustering performance without smoothing on DLPFC dataset across foundation models.

We then independently apply K-Means clustering to all feature vectors $\{\mathbf{z}_i\}_{i=1}^N$ within each slide. The number of clusters is fixed to $k = 7$, corresponding to the seven anatomical layers of the human DLPFC. The clustering procedure is initialized with $n_{\text{init}} = 50$ random seeds, a maximum of 500 iterations, and uses Euclidean distance as the similarity measure. This process outputs the initial cluster label for each spot, denoted as $\hat{y}_i^{\text{raw}}$. We then introduce a spatial label smoothing mechanism to improve spatial consistency. Specifically, for each spot, we use its two-dimensional coordinates $(x_i, y_i)$ to construct a $k$-nearest neighbor graph (default $k = 5$) and refine the initial

cluster labels through majority voting. At each smoothing iteration, the update rule is defined as:

$$\hat{y}_i^{(t+1)} = \begin{cases} \arg\max_c \ \#\{j \in \mathcal{N}_k(i) : \hat{y}_j^{(t)} = c\}, \\ \hat{y}_i^{(t)} \end{cases} \tag{15}$$

Here, $\mathcal{N}_k(i)$ denotes the $k$ nearest neighbors of spot $i$ in spatial coordinates. The smoothing procedure is repeated for $T = 1$ iteration by default, producing the final smoothed label $.\hat{y}_i^{\text{sm}}$.

### C.2.2 FEW-SHOT PREDICTION TASK

For slides with available anatomical layer annotations, we adopt the Adjusted Rand Index (ARI) and Normalized Mutual Information (NMI) as evaluation metrics, measuring the structural consistency between predicted labels and ground truth. Both raw clustering labels $\hat{y}^{\text{raw}}$ and smoothed labels $\hat{y}^{\text{sm}}$ are evaluated, and the metrics are computed for each slide and then averaged across the dataset.

We followed the transfer learning paradigm introduced in ST-Net and adopted a few-shot prediction setting to evaluate the transferability and generalization of models when applied to new datasets with minimal fine-tuning. Specifically, we treated the image encoder of each pretrained foundation model as the backbone and appended a fully connected regression head to predict gene expression.

During training, we fixed the random seed and randomly sampled different proportions of spots from the training set (10%, 25%, 50%) as few-shot samples, then examined how the performance on the full test set varied with the sampling ratio. To ensure fairness, all models were trained under the same optimization configuration. We used an SGD optimizer with a momentum of 0.9 and a weight decay of $10^{-4}$. The learning rate was set to $5 \times 10^{-5}$ with a batch size of 256, and each model was fine-tuned for 30 epochs on each dataset.

Moreover, we performed sample-level four-fold cross-validation on each dataset to mitigate biases caused by data partitioning. The average results across folds were reported as the final performance metrics. As shown in Table 2, we present the mean values of three widely used evaluation measures, MSE, MAE and PCC, to comprehensively assess the accuracy of the models.

| Category | Setting | ARI | NMI |
|----------|---------|-----|-----|
| Functional Block | w.o TF-IDF | 0.1861 | 0.3035 |
| | w.o Regional Align | 0.2262 | 0.3252 |
| | HiBio-ST (Ours) | 0.2640 | 0.3776 |
| Pathway Ratio | $\tau = 0.1$ | 0.2288 | 0.3458 |
| | $\tau = 0.2$ | 0.2640 | 0.3776 |
| | $\tau = 0.5$ | 0.2399 | 0.3703 |
| Loss Weights | $\lambda_2 = 0, \lambda_3 = 0$ | 0.2092 | 0.3112 |
| | $\lambda_2 = 0.25, \lambda_3 = 0.1$ | 0.2306 | 0.3512 |
| | $\lambda_2 = 0.25, \lambda_3 = 0.25$ | 0.2640 | 0.3776 |
| | $\lambda_2 = 0.5, \lambda_3 = 0.5$ | 0.2678 | 0.3871 |
| HVG Selection | Token Length=250 | 0.1573 | 0.2726 |

Table 7: Ablation study of zero-shot clustering performance on DLPFC dataset.

### C.2.3 GENE EXPRESSION PREDICTION TASK

We selected two mainstream training paradigms for directly inferring gene expression from histology images: regression-based baselines and retrieval-based approaches with contrastive learning. The regression-based methods directly predict gene expression from input images, including ST-Net (He et al., 2020), EGN (Yang et al., 2023), HisToGene (Pang et al., 2021), His2ST (Zeng et al., 2022), and TRIPLEX (Chung et al., 2024). In contrast, retrieval-based approaches leverage contrastive learning to align image and expression representations, with representative models such as BLEEP (Xie et al., 2024), mlxExpST (Min et al., 2024), OpenCLIP (Ilharco et al., 2021), and OmiCLIP (Chen et al., 2025). For evaluation, we employed the same datasets used in the few-shot prediction experiments and performed four-fold cross-validation, reporting the average results across folds as the final metrics.

For the regression-based baselines, we followed the original training configurations specified in their respective works and trained each model until convergence. To ensure fairness and consistency,

we applied several adjustments. For HisToGene and His2ST, which were originally designed for $112 \times 112$ pixel spot images, we resized all patches to $224 \times 224$ pixels to unify input dimensions. For EGN, we replaced the original GAN-based encoder with a pretrained ViT-B-16 backbone, thereby standardizing feature extraction across regression-based models. In line with the original EGN implementation, we further selected the eight most similar exemplars for each spot during training.

For the retrieval-based approaches, predictions were generated by aggregating the top 100 most similar spots, identified according to the similarity between image/text embeddings and gene expression profiles. We largely followed the original settings of each method, with minor adjustments for fairness. Specifically, for BLEEP, mlxExpST, and OpenCLIP, we employed ViT-B-16 as the image encoder to ensure a consistent backbone across methods.

Finally, to ensure fairness, all models were trained with the same optimization setup. We employed SGD with momentum fixed at 0.9 and weight decay of $10^{-4}$. The training used a learning rate of $5 \times 10^{-5}$ and a batch size of 256. For each dataset, fine-tuning was conducted for 30 epochs.

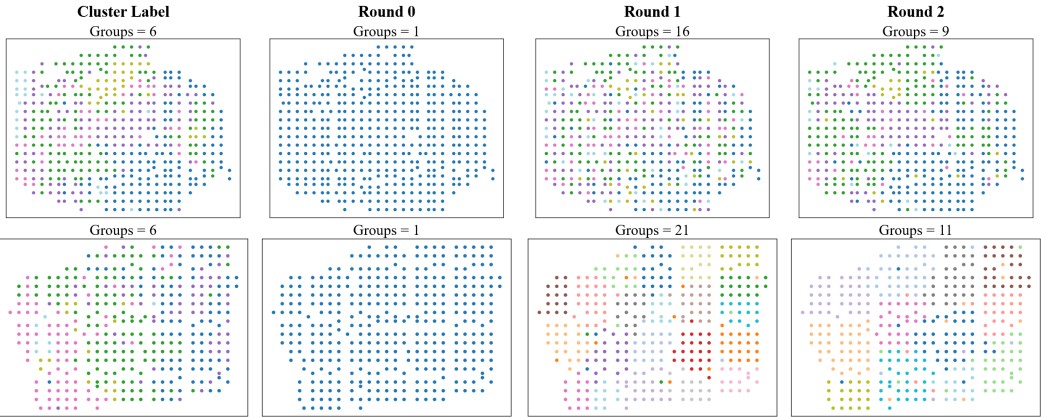

Figure 9: Visualization of the progressive refinement of regional identities alignment for each spatial spot, illustrating how initially uniform assignments evolve into coherent tissue regions driven by both gene signals and visual features.

### C.2.4 IMAGE-TO-ST SENTENCE RETRIEVAL TASK

In the Image-to-ST sentence retrieval task, we standardized the input to be a sentence composed of 250 gene names. For OmiClip (Chen et al., 2025) and PLIP (Huang et al., 2023), we retained their original tokenizers to encode the gene sentences. However, both models are designed with a maximum input length of 77 tokens, which inevitably truncates the sentences and prevents the full set of genes from being represented. Moreover, since both models adopt byte-pair encoding (BPE), a single gene name is typically split into 2–4 subword tokens. As a result, although the input sentence contains 250 genes, only about 20–40 genes can be effectively encoded and learned, substantially limiting the amount of information available for retrieval and compressing the effective length of the gene sentence.

In contrast, for OpenCLIP (Ilharco et al., 2021), we employed a text transformer with a maximum sequence length of 280 tokens and adopted the same strategy as our proposed method, treating each gene name as an individual token. This design avoids the over-segmentation problem inherent to BPE and ensures that most, if not all, of the genes can be directly represented and aligned at the token level. Consequently, OpenCLIP is able to preserve gene-level semantics in the retrieval task.

For optimization, we applied a unified fine-tuning configuration across all models to ensure fairness. Specifically, all experiments were trained with the InfoCE loss using the SGD optimizer with a momentum of 0.9, a weight decay of $10^{-4}$, a learning rate of $5 \times 10^{-5}$, and a batch size of 256. Each dataset was fine-tuned for 10 epochs, under which OmiClip, PLIP, OpenCLIP, and our method were evaluated consistently.

# D  ADDITIONAL EXPERIMENTAL RESULTS

## D.1  REGION IDENTIFIER UPDATES.

We conducted additional pretraining experiments on HER2 samples containing expert-annotated spatial clusters (6 groups) to visualize the progressive refinement of regional identity alignment for each spot. As shown in Figure 9, at the initial stage, all spots are initialized to a single group, reflecting the absence of spatially meaningful structure. After several updates, the number of region groups gradually decreases, and adjacent spots begin to aggregate into coherent domains. This refinement emerges from the combined signals of ST gene similarity and visual feature proximity, which guide nearby spots toward consistent regional identities. During this process, the spatial layout becomes more structured and less chaotic, forming biologically interpretable tissue regions.

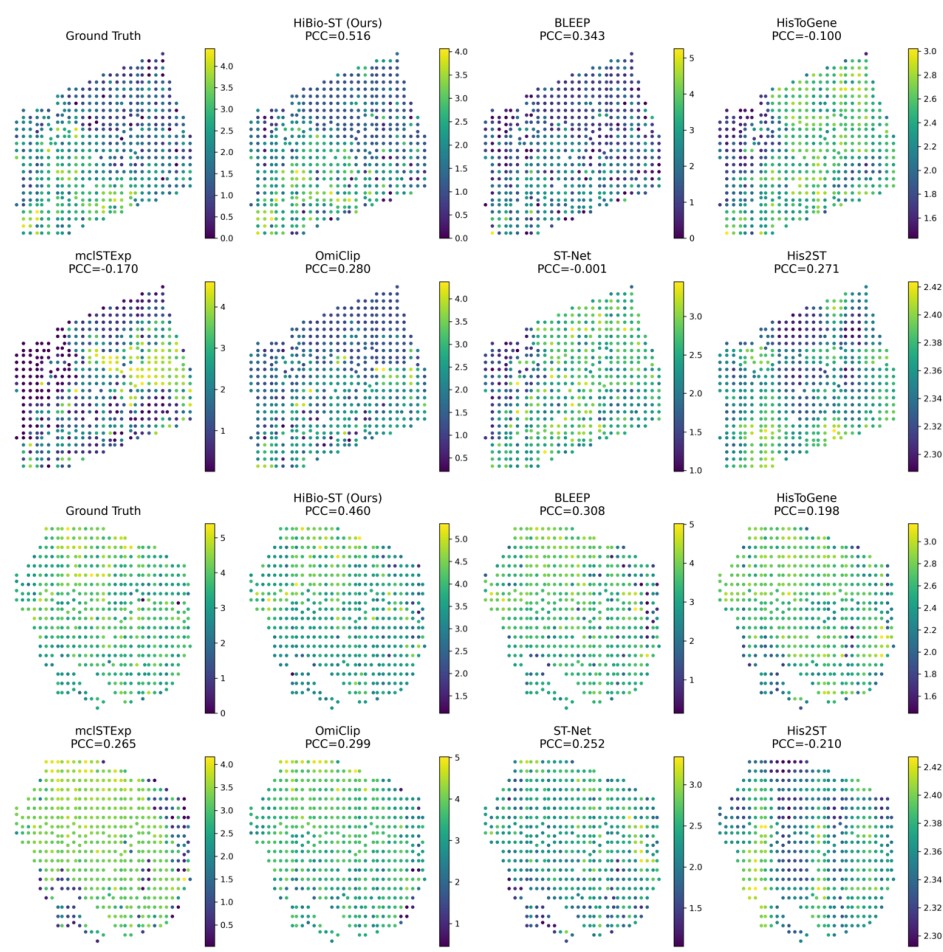

Figure 10: Additional visualization of the predicted spatial expression distribution of the cancer-associated gene IGLL5 on WSIs.

## D.2  ADDITIONAL DOWNSTREAM RESULTS

In this section, we present supplementary experimental results for each downstream task. These results are not included in the main text because of space limitations.

**Quantitative Analysis and Ablation Study on Zero-shot Layer Clustering.** As shown in Table 7 and Figure 7, we systematically evaluated the zero-shot hierarchical clustering performance ablation

settings on the DLPFC dataset. The table reports the average ARI and NMI scores across selected DLPFC slides. We also visualize the cluster performance on HER2 dataset in Figure 8. Overall, baseline methods such as OmiCLIP, CONCH, and UNI exhibited limited clustering capability. The ablation studies further highlighted the importance of key modules: after progressively integrating all functional blocks, the ARI and NMI of HiBio-ST improved from 0.1861 and 0.3035 to 0.2640 and 0.3776, respectively. Meanwhile, the visualizations in Figure 7 corroborate these findings, with the incorporation of the regional alignment module, our model achieves superior alignment with the laminar architecture.

**Few-shot Prediction Results on HER2 Dataset.** Figure 2 presents the few-shot gene expression prediction performance on the HER2 dataset. Consistent with the trends observed on the Breast Cancer and Kidney cohorts, HiBio-ST demonstrates consistently superior generalization across all sampling ratios. The advantage becomes especially pronounced under the 10% data setting, where the limited number of available spots greatly challenges the ability of models to learn stable and reliable mappings between histology and gene expression. These results further underscore the strong transferability and generalization capacity of our approach, which remains effective even when only minimal fine-tuning data is accessible.

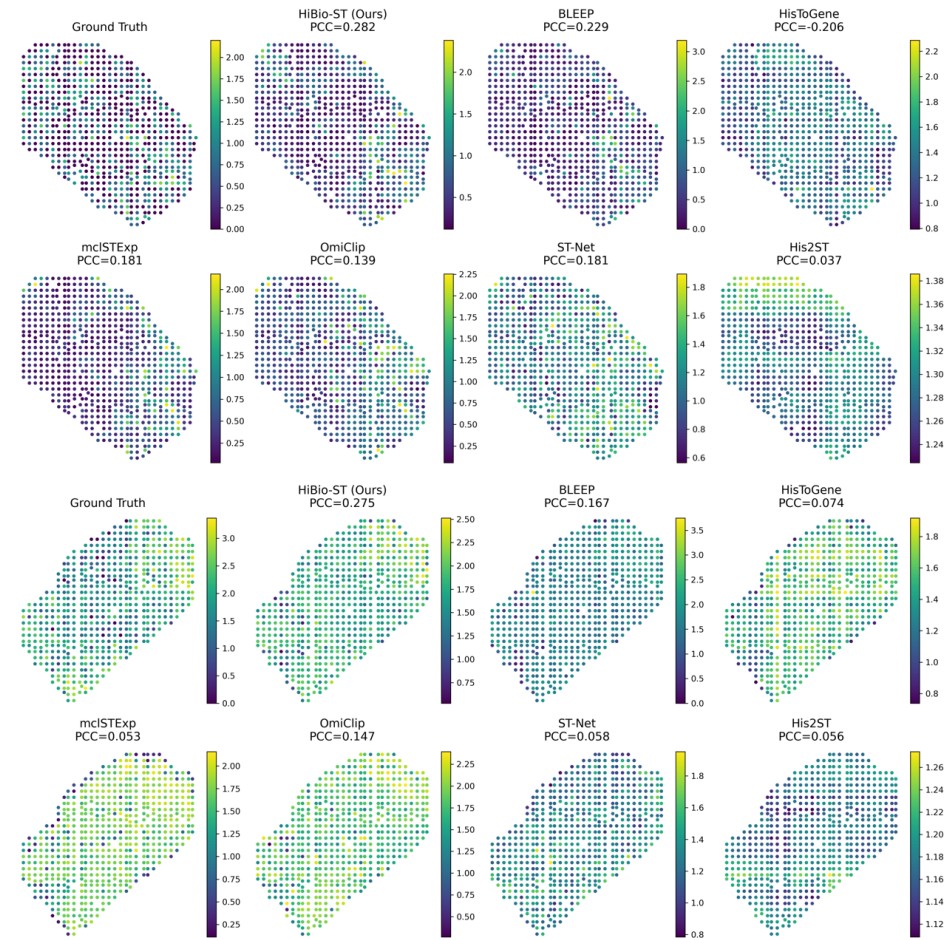

Figure 11: Visualization of the predicted spatial expression distribution of cancer-related gene IFI27.

**Cancer-related Gene Expression Prediction.** In addition to the tumor-related gene IGLL5, we further selected two genes that are highly relevant to breast cancer, MGP and IFI27. IFI27, a proto-typical interferon-stimulated gene, is frequently upregulated in breast cancer and has been shown to be associated with poorer overall survival (Cervantes-Badillo et al., 2020; Choi et al., 2015). MGP, on the other hand, is a key extracellular matrix regulator whose elevated mRNA levels in breast cancer patients correlate with unfavorable prognosis, and it also demonstrates high detection sensi-

| Resolution | Model | Breast Cancer-HD | | | CRC-HD | | |
|---|---|---|---|---|---|---|---|
| | | MSE | MAE | PCC | MSE | MAE | PCC |
| 8um | ST-Net | 0.3829 | 0.3591 | 0.0514 | 0.2283 | 0.2768 | 0.0821 |
| | His2ST | 0.3577 | 0.3712 | 0.0628 | 0.2416 | 0.2913 | 0.0104 |
| | HisToGene | 0.4025 | 0.3867 | 0.0098 | 0.2145 | 0.2891 | 0.0477 |
| | BLEEP | 0.2984 | 0.3415 | 0.1097 | 0.1742 | 0.2194 | 0.1026 |
| | mclSTExp | 0.2831 | 0.3619 | 0.0833 | 0.1529 | 0.2077 | 0.1214 |
| | OmiCLIP | 0.4192 | 0.3278 | 0.1394 | 0.1728 | 0.2361 | 0.1499 |
| | HiBio-ST (Ours) | 0.2498 | 0.2986 | 0.1139 | 0.1383 | 0.1724 | 0.1715 |
| 16um | ST-Net | 0.9723 | 0.7422 | 0.1209 | 0.5727 | 0.4938 | 0.1045 |
| | His2ST | 1.1475 | 0.7936 | 0.1375 | 0.6513 | 0.5179 | 0.0427 |
| | HisToGene | 0.9016 | 0.7018 | 0.1094 | 0.5052 | 0.5026 | 0.0968 |
| | BLEEP | 0.8154 | 0.6613 | 0.1691 | 0.4957 | 0.4821 | 0.1023 |
| | mclSTExp | 0.8577 | 0.6721 | 0.1366 | 0.4624 | 0.4498 | 0.1182 |
| | OmiCLIP | 0.7889 | 0.6119 | 0.1241 | 0.4811 | 0.4685 | 0.1815 |
| | HiBio-ST (Ours) | 0.7138 | 0.6397 | 0.1674 | 0.4202 | 0.4095 | 0.1628 |

Table 8: **Quantitative comparisons on gene expression prediction on high resolution datasets.**

tivity across multiple breast cancer subtypes (Du et al., 2022; Yoshimura et al., 2009). We further visualized the predicted spatial distributions of these two genes on WSIs in Figure 11 and Figure 12, thereby highlighting the model's interpretability at clinically critical molecular levels. In addition, Figure 10 presents the predicted expression distribution of IGLL5 as a complementary analysis.

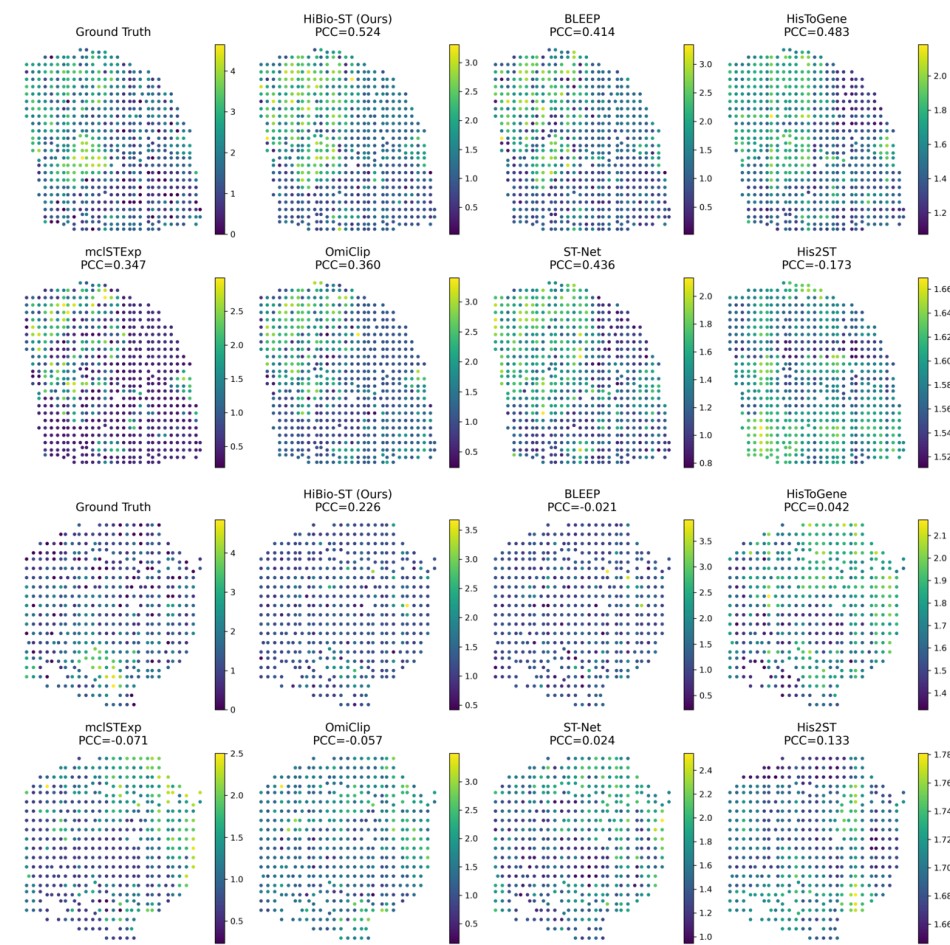

Figure 12: Visualization of the predicted spatial expression distribution of cancer-related gene MGP.

**Comparison with HVG selection.** To evaluate the effectiveness of our gene selection strategy, we conducted additional experiments using the High Variance Gene (HVG) selection method under the

| Resolution | Model | Token Length | CRC-HD | | |
| --- | --- | --- | --- | --- | --- |
| | | | Recall@1% | Recall@5% | Recall@10% |
| 8um | Random | - | 0.736 | 4.279 | 7.863 |
| | Openclip | 250 | 1.431 | 6.936 | 13.457 |
| | PLIP | 76 | 1.897 | 9.358 | 17.499 |
| | Omiclip | 76 | 2.376 | 11.637 | 21.305 |
| | Conch | 128 | 2.103 | 7.683 | 16.917 |
| | HiBio-ST (Ours) | 250 | 2.915 | 14.299 | 24.295 |
| 16um | Random | - | 0.858 | 6.103 | 9.755 |
| | Openclip | 250 | 1.621 | 8.570 | 18.354 |
| | PLIP | 76 | 2.203 | 12.841 | 22.792 |
| | Omiclip | 76 | 2.972 | 16.708 | 30.305 |
| | Conch | 128 | 1.988 | 9.795 | 24.329 |
| | HiBio-ST (Ours) | 250 | 3.332 | 21.089 | 28.356 |

Table 9: **Image-to-ST sentence retrieval performance on high resolution dataset.**

same experimental settings. Specifically, we computed gene variance at the slide level and selected high-variance genes within each slide to construct a shared HVG gene sentence, meaning that all spots from the same slide use an identical HVG vocabulary. This HVG-based strategy follows exactly the same pretraining and downstream evaluation procedures as our original method. The corresponding results have been added to Table 4,Table 5, Table 7, and Figure 7.

The experimental results show that the model exhibits a clear performance drop on almost all downstream tasks under this setting, with most results even falling below those obtained using the Top-gene selection strategy. We attribute this phenomenon to the fact that HVG selection relies on slide-level variance statistics, thus providing only bulk-level shared gene information and failing to capture the fine-grained, spot-level biological semantics. In contrast, both TF-IDF and Top-gene strategies operate on the expression distribution of individual spots, thereby preserving spatial heterogeneity and localized distribution patterns. As a result, these two spot-specific strategies demonstrate greater stability and superiority in spot-level cross-modal tasks. Thus, we validate that TF-IDF is a more suitable and effective strategy for constructing gene sentences and establishing a unified semantic vocabulary for pretraining.

**Performance on High Resolution dataset.** We further selected two high-resolution Visium HD datasets to evaluate the model's performance under high-resolution spatial transcriptomics conditions, including the Breast Cancer HD dataset (3 slides) Nagendran et al. (2023) and the CRC HD dataset (5 slides) Oliveira et al. (2025). To ensure consistency with our original visual input design, we cropped 224×224 px image patches centered at each bin, and conducted experiments under two resolution settings: 8 $\mu$m and 16 $\mu$m. In terms of experimental protocol, we used 3-fold cross-validation for the Breast Cancer HD dataset and 5-fold cross-validation for the CRC dataset. Table 8 and Table 9 present the results of various methods on these high-resolution datasets for the gene expression prediction task and the image-to-ST-sentence retrieval task, respectively, demonstrating that the proposed model maintains stable and superior effectiveness under multi-resolution settings.

# E CLARIFICATION OF GENERATIVE AI

This paper was polished by using the AI tool (ChatGPT) with the prompt "rewrite."