# OpenReview forum: "HiBio-ST: A Hierarchical Multimodal Foundation Model with Biological Prior Anchors for Spatial Transcriptomics"
_ICLR.cc/2026/Conference — Submitted to ICLR 2026_

### Official Review · Reviewer_8X2p · 2025-11-01

**Soundness:** 2
**Presentation:** 3
**Contribution:** 2
**Rating:** 4
**Confidence:** 5

**Summary:**

The authors propose HiBio-ST, a hierarchical multimodal foundation model for ST that: (1) reweights genes via TF-IDF to emphasize “keyword” genes and use ranked gene names as gene sentence , (2) provides a way to construct pathway priors using curated gene-set “anchors,” and (3) adds region-aware clustering to encourage meso-scale spatial coherence. While the results show improved compared to several SOTA methods, I am not fully convinced by the results that the gene ranking methods are more efficient and powerful than some current methods like spatial variable gene detection. Additional experiments are needed to support the claim.

**Strengths:**

The authors provide comprehensive experiments for various tasks. The idea of use external information like pathway gene sets is interesting and potential useful.

**Weaknesses:**

Additional experiments and discussion need to be added for a comprehensive comparison with SOTA methods.

1. The authors claim that "the high-expression-only strategy overlooks spot-specific signals and instead favors ubiquitous housekeeping genes". There are no evidence in the paper that support this claim, the authors should provide examples in certain datasets comparing the genes highlighted by TF-IDF or pathway and HVGs.
2. In practice, in a lot of spatial clustering methods, people use spatial variable genes (SVGs) instead of HVGs. How does TF-IDF ranked genes compare with SVGs? The author could consider benchmark against SVGs extracted using SPARK, Moran's I, and nnSVG.
3. It has been discussed in several papers (Genept, SpatialAgent) that the gene could be ranked in a better way than just using HVGs or mean/max gene expression. Please add discussion about them if they are not discussed. How does TF-IDF or pathway informed gene importance/weights compared to the gene importance proposed in Genept and SpatialAgent?
4. There are many overlapping pathways relevant to a given tissue (especially for tumors: immune, tumor invasion, and tumor–immune interaction). Please explain how you selected which pathways to include and whether this choice generalizes across tissues. And for healthy tissue like DLPFC, how should users choose pathway? Please add experiment about pathway informed genes for the DLPFC analysis.
5. The authors should add more comparison with recent multimodal or fine-tuned baselines, such as: fine-tuned CLIP (e.g., from STimage-1K4M), fine-tuned CONCH (e.g., from HEST-1K), and several other multi-modal framework, for example, UMPIRE, and STPath frameworks. These comparisons are essential to position your method against current SOTA.

**Questions:**

See Weaknesses for major questions.

Minor questions:
1. For clustering task, the authors only evaluate zero-shot clustering task on DLPFC but not HER2+ breast cancer, while they evaluate linear probing on the HER2+ but not DLPFC. Please include the result for both datasets.

---

> ### Author Response · Authors · 2025-11-25
>
> **1. Genes with TF-IDF Selection:** Many essential genes exhibit low expression levels and yet are functionally indispensable [1, 2]. Many essential genes exhibit low expression levels. For example, CST6 and BRMS1 are lowly expressed tumour suppressor genes in breast cancer [3]. In spatial transcriptomics, such genes typically show weak global expression and only modest down-regulation in specific regions. These genes are often overlooked by conventional selection strategies: (1) high-expression ranking favors globally abundant genes, while (2) HVG selection relies on slide-level variance and tends to prioritize genes with large overall dynamic ranges rather than those that are selectively expressed in a small subset of spots. In contrast, TF-IDF does not depend on absolute expression or global variance, but instead evaluates gene importance based on the cross-spot frequency distribution, allowing it to capture these locally enriched, biologically relevant signals that traditional methods fail to detect.
>
> > [1] Chen H, Zhang Z, Jiang S, et al. New insights on human essential genes based on integrated analysis and the construction of the HEGIAP web-based platform[J]. Briefings in bioinformatics, 2020, 21(4): 1397-1410.
> >
> > [2] Chen L, Pan X Y, Zhang Y H, et al. Classification of widely and rarely expressed genes with recurrent neural network[J]. Computational and structural biotechnology journal, 2019, 17: 49-60.
> >
> > [3] Liang, Ying, et al. "Reactivation of tumour suppressor in breast cancer by enhancer switching through NamiRNA network." *Nucleic acids research* 49.15 (2021): 8556-8572.
>
> ---
>
> **2. Zero-shot Clustering:** SVGs are primarily used in **gene expression–based spatial clustering**. Such methods inherently rely on costly ST expression matrices and require statistical tools such as SPARK, Moran’s I, or nnSVG for computation. In contrast, our work focuses on **purely vision-driven spatial clustering**, investigating whether meaningful spatial structure can be inferred solely from pretrained histology representations without using any costly ST data. This task targets practical scenarios where RNA sequencing is difficult to obtain, thereby avoiding the high cost of expression acquisition and evaluating the potential of visual foundation models for spatial tissue analysis.
>
> Therefore, our intention is not to replace SVGs within an expression-based clustering framework, but rather to provide an alternative clustering approach that does not rely on expression data. Under this task formulation, we fairly compare different visual foundation models by evaluating their representation ability in a purely image-based zero-shot clustering setting.
>
> ---
>
> **3. Gene selection strategy in related works:** We appreciate the reviewer’s point regarding related work. We have now added a discussion of GenePT and SpatialAgent in Supplementary *Section B Related Work* (Line 922) of the revised manuscript and clarified how they differ from our approach.
>
> > Our proposed TF-IDF with pathway prior aims to construct a shared gene vocabulary across samples and cohorts for vision–transcriptome pretraining. The goal is to obtain a stable and generalizable gene token space that allows different slides, tissues, and datasets to encode into a unified textual representation, thereby enabling large-scale pretraining. In contrast:
> >
> > (1)  SpatialAgent serves broader spatial biology workflows, particularly task-specific gene panel design. Its notion of gene importance reflects task objectives rather than building a transferable pretraining vocabulary shared across datasets. Thus, its goal differs fundamentally from ours.
> >
> > (2)  GenePT, similar to OmiCLIP, constructs cell sentences by ranking gene names based on expression levels. Such sequences tend to be dominated by top-expressed but high-abundance housekeeping genes, and may underrepresent low-to-moderate expression signals that carry spatial specificity.
>
> ---

---

> ### Author Response · Authors · 2025-11-25
>
> **4. Pathway selection:** We appreciate the reviewer’s comment regarding pathways selection.
>
> (1)  Here, we determine positive semantic anchors through the overlap between spot-specific expressed genes and pathway gene sets, where pathways with higher overlap serve as soft positives in our image–omics alignment. Following other reviewers’ suggestions, we further evaluated different pathway-overlap thresholds and their impact on pretraining. The results, summarized in the revised manuscript and the Table 4, 5 and Table 7, demonstrate consistent gains, supporting the robustness of our overlap-based pathway selection strategy.
>
> (2)  **Pathways are not limited to disease-related mechanisms.** Public resources such as KEGG include numerous fundamental physiological processes (e.g., metabolic regulation, synaptic signaling), which remain active and observable in healthy tissues such as the DLPFC [1-3].
>
> Furthermore, our framework does not require a pathway to always be a “positive” match. In spots where no pathway is detected, these non-overlap pathways naturally become negative semantic anchors, **providing a negative structured prior** that helps define separable functional boundaries in the embedding space. **Thus, a pathway does not need to be explicitly confirmed as “active” to be informative; instead, it functions jointly as a positive or negative semantic anchor to constrain the image-gene alignment.**
>
> > [1] Kanehisa M, Goto S, Sato Y, et al. KEGG for integration and interpretation of large-scale molecular data sets[J]. Nucleic acids research, 2012, 40(D1): D109-D114.
> > [2] Kanehisa M. Toward understanding the origin and evolution of cellular organisms[J]. Protein science, 2019, 28(11): 1947-1951.
> > [3] Kanehisa M, Furumichi M, Sato Y, et al. KEGG: integrating viruses and cellular organisms[J]. Nucleic acids research, 2021, 49(D1): D545-D551.
>
> ---
>
> **5. Additional baseline comparison:** We appreciate the reviewer’s suggestion on current SOTA baselines. Following your suggestion, we have added additional comparisons with recent multimodal baselines, including UMPIRE and STPath, under the same evaluation setup. The results are now incorporated in the revised manuscript and reported in Figure 2 & Figure8, and Table 1 & Table 6.
>
> ---
>
>
> **6. Additional experiments on HER2 and DLPFC:** We appreciate the reviewer’s suggestion. In the revised version, we have added linear probing results on the DLPFC dataset as well as zero-shot clustering performance on the HER2+ dataset. These results are now included in the revised manuscript and presented in Figure 2 & Figure8, and Table 1.

---

> > ### Comment · Reviewer_8X2p · 2025-11-26
> >
> > Thanks for all the response and revision.
> >
> > For 2: Regarding the SVG comparison you respond in zero-shot clustering, I am referring to use SVG rank when you train your multi-modal foundation model.
> >
> > For 4: Thanks for the additional explanation. I might have missed, but could you provide guideline on how to select the pathway to use for a given dataset?

---

> > > ### Author Response · Authors · 2025-11-26
> > >
> > > Thank you for recognizing our efforts. Here, we would like to address your remaining concerns and questions:
> > >
> > > ---
> > >
> > > **1. SVG-based Ranking:** Thank you for clarification and insightful comment regarding gene sentence construction. We agree that SVGs or HVGs provide meaningful spatial information within a single slide. However, directly using SVG ranking to construct gene sentences for multi-modal foundation model training has two critical limitations:
> > >
> > > **(1) SVGs cannot provide spot-level discriminability within a slide.** In our framework, gene sentences are constructed based on **selected gene names.** In contrast, SVG rankings are computed from the entire expression matrix of one slide; therefore, **all spots within the same slide (or batch) share the same ranked gene list rather than having spot-specific gene selections.** As a result, **using SVGs would assign identical gene names to all spots from the same slide,** making it impossible to differentiate instance-level patches during training, and forcing the model into a slide-level bulk supervision, rather than learning fine-grained spot-level image–omics correspondences.
> > >
> > > **(2) Gene readouts could be unstable across cohorts and dominated by technical variability.** If SVG or HVG rankings were used to construct gene sentences, raw expression values must be involved. Thus, these rankings are **highly sensitive to technical confounders across cohorts**, including sequencing depth, capture efficiency, tissue handling, and batch effects. Consequently, the gene selection would reflect dataset-specific technical variations rather than reproducible biological semantics, causing the model to **learn technical noise instead of transferable image–omics semantics.**
> > >
> > > To avoid these pitfalls, therefore, we adopt TF-IDF gene selection with a **unified gene vocabulary** for spot-specific gene selection, where gene names themselves act as a stable semantic space.
> > >
> > > ---
> > >
> > > **2. Pathway selection:** We appreciate your comments on pathway selection. Here, we would like to clarify that our method does not rely on any manual selection of pathways based on a particular dataset or disease context, nor does it conduct pathway filtering at the dataset level. **Instead, we use KEGG as a universal functional dictionary, and the pathways engaged during training are automatically activated at the spot level, driven solely by each spot’s expression profile, rather than being predefined by researchers.** The pathway activation mechanism is summarized in the manuscript (Line 151–172, Line 217–225) and as follows :
> > >
> > > 1. Each spot first obtains a representative gene subset $G_S$ via TF-IDF filtering, capturing its localized expression pattern.
> > >
> > > 2. For a given pathway $P$ with gene set $G_P$, the pathway is considered functionally relevant to that spot only when the normalized overlap between $G_P$ and $G_S$ exceeds a threshold $\tau$. Relevant pathways are then used for cross-modal alignment.
> > >
> > > 3. Pathways satisfying this threshold are treated as positive candidates and aligned using soft overlap weights $(\omega_j)$; in contrast, pathways not meeting this relevance condition are automatically treated as negative samples and suppressed through contrastive training.
> > >
> > > In response to other reviewers’ suggestions, we additionally report a performance comparison under different values of the pathway overlap ratio $\tau$. These results are summarized in **Tables 4, 5, and 7**, and a detailed analysis has been added in Section *Pathway Overlap Ratio* (Line 481).
> > >
> > > > **Pathway Overlap Ratio.** We evaluate two configurations (τ=0.1 and τ=0.5) in comparison with our default setting (τ=0.2) to better understand the influence of hyperparameters in our framework, with results summarized in Tables 4, 5, and 7. The results indicate that a low overlap ratio leads to noticeable performance degradation. This occurs because reducing τ introduces an excessive number of positive pathway pairs, thereby increasing the probability of incorporating noisy or biologically irrelevant pathways as anchors. In contrast, when τ becomes overly strict, the performance either plateaus or slightly decreases, as the number of valid pathway anchors becomes too limited to provide sufficient biological supervision during contrastive alignment. Thus, a moderate overlap ratio yields the best trade-off between eliminating noisy anchors and retaining informative and biologically meaningful pathway anchors.

---

> > > > ### Comment · Reviewer_8X2p · 2025-11-28
> > > >
> > > > Thanks for the additional clarification. I think the paper has been improved during the rebuttal and decide to raise the score to 6.

---

> > > > > ### Author Response · Authors · 2025-11-30
> > > > >
> > > > > Thank you for your positive feedback and for acknowledging the improvements in our manuscript. We sincerely appreciate your constructive review and valuable suggestions.

---

### Official Review · Reviewer_hemv · 2025-11-01

**Soundness:** 2
**Presentation:** 3
**Contribution:** 2
**Rating:** 4
**Confidence:** 4

**Summary:**

This paper proposes HiBio-ST, a multimodal foundation model that integrates pathology images and spatial transcriptomics (ST) data for diverse ST analyses. It aims to overcome the poor generalizability of task-specific models and the limitations of existing multi-modal foundation models, which often (1) overlook spot-specific signals and (2) fail to capture the microenvironmental context in spot-patch pairs.

To address these issues, HiBio-ST introduces three components:
1. TF-IDF for Gene Selection: Selects genes based on relative scarcity rather than absolute expression, capturing important spot-specific signals.
2. Pathway-guided integration in spot-patch pairs: Incorporates pathway information to apply global biological constraints to spot-patch pairs.
3. Region-Aware Hierarchical Clustering: Uses Leiden clustering enhanced by edge weights derived from both spatial distance and feature cosine distance to better reflect biological structures.

These components are aligned using a contrastive loss objective. Applies contrastive loss objectives to simultaneously align visual and gene embeddings at the local (spot), functional (pathway), and meso-scale (region) levels.

**Strengths:**

* The paper clearly explains its key motivation. (e.g., needs for foundation model in this ST domain, limitation of gene selection)
* The paper is well-written and easy-to-follow.

**Weaknesses:**

* While the claim that a foundation model is needed for better generalizability compared to task-specific models is valid, the zero-shot clustering performance presented to substantiate this is very poor. Specifically, the performance is significantly lower than that of representative task-specific models [1, 2] that use only spatial transcriptomics. Consequently, the proposed method does not appear to support its stated motivation.
* The methodology is considered incremental, as it primarily combines existing TF-IDF gene selection from the single-cell domain [3] with community detection [4, 5].
* The method section lacks a clear explanation of how the individual components of HiBio-ST are integrated, particularly in Section 2.4. For instance, it is unclear how the pathway prototype representation, which is an aggregation of gene embeddings, is applied in Equation (11).
---
[1] Deciphering spatial domains from spatially resolved transcriptomics with an adaptive graph attention auto-encoder. Nature communications. 2022.

[2] Global Context-aware Representation Learning for Spatially Resolved Transcriptomics. ICML. 2025.

[3] Single cell RNA-seq data clustering using TF-IDF based methods. BMC genomics. 2018.

[4] Modularity and community structure in networks. PNAS. 2006.

[5] From Louvain to Leiden: guaranteeing well-connected communities. Scientific Reports. 2019.

**Questions:**

* There is no definition for $G_s$. Is it the set of expressed genes within a specific spot?
* The paper argues that a foundation model is necessary for generalizability, but the reported zero-shot performance is poor. Even if it is inferior to task-specific models, shouldn't the performance be at a comparable level? Is a direct comparison with these task-specific models available?
* Alternatively, are there other ways to demonstrate generalizability from a different perspective?
* With technological advancements, newer, higher-resolution spatial transcriptomics technologies like Xenium (which is part of HEST-1k) are available. How does HiBio-ST perform on this type of data?
* TF-IDF-based gene selection performs better than existing HVG (highly variable genes) and SVG (spatially variable genes) selection?
* How about parameter sizes and time complexity for pretraining and inference comparing with baselines?

---

> ### Author Response · Authors · 2025-11-25
>
> **1. Definition for Gs:** We thank the reviewer for pointing out this omission. In our framework, $G_s$ refers to the spot-specific gene set selected by the TF-IDF–based filtering strategy. We have added this definition to the revised manuscript for clarity (Line 154).
>
> ---
>
> **2. Zero-shot clustering task:** We appreciate the reviewer’s question. Here we clarify the fundamental difference in task formulation and evaluation settings.
>
> Our task is **purely vision-driven.** The model only uses image embeddings to perform clustering or retrieval, without accessing any gene expression values during inference. This is an intentional design motivated by the practical constraints of spatial transcriptomics, where gene expression measurement is extremely expensive, whereas histology images are readily available. This differs substantially from existing task-specific clustering methods, which require gene expression profiles with access to molecular features. As a result, the performance scales and upper bounds between these two categories of methods are inherently different and cannot be compared in a one-to-one manner.
>
> To ensure fairness, we only compare with other vision-based foundation models under the same setting, and our method consistently achieves superior performance in this comparable scenario.
>
> ---
>
> **3. Evaluation for generalizability:** In addition to the zero-shot clustering task, we also evaluate the model’s generalizability from another perspective. Specifically, we include a few-shot gene expression prediction task, with results summarized in Figure. 2. In this setting, we fine-tune the model using only a very limited number of labeled ST samples and examine how the performance improves under scarce supervision, providing an additional assessment of generalizability. This few-shot evaluation directly reflects the model’s adaptability in realistic low-resource scenarios [1,2].
>
> > [1] Radford A, Kim J W, Hallacy C, et al. Learning transferable visual models from natural language supervision[C]//International conference on machine learning. PmLR, 2021: 8748-8763.
> >
> > [2] Shakeri F, Huang Y, Silva-Rodríguez J, et al. Few-shot adaptation of medical vision-language models[C]//International Conference on Medical Image Computing and Computer-Assisted Intervention. Cham: Springer Nature Switzerland, 2024: 553-563.
>
> ---
>
> **4. High resolution datasets:** We appreciate the reviewer’s comment. Platforms such as Visium HD and Xenium operate at spatial scales fundamentally different from traditional Visium ST. For example, Visium HD provides 2-8 µm bins, whereas Visium ST spots are approximately 55-100 µm. An 8 µm bin corresponds to roughly a 32 px × 32 px image patch, far smaller than the ~224 px × 224 px patches used for Visium ST in our model. High-resolution platforms also exhibit substantially different detection rates, spatial patterns, and noise characteristics compared with lower-resolution ST [1, 2]. **These differences in spatial resolution, sampling density, and gene-expression distribution indicate that the two data types are not directly comparable.**
>
> **Given these distributional discrepancies, the limited scale of available high-resolution datasets, and our goal to maintain cross-platform consistency while avoiding additional noise, we restrict our training to Visium/Visium ST data.** We believe this choice leads to more stable and reliable generalization under the current data regime. As larger high-resolution ST datasets become available, extending our framework to these platforms will be an important direction for future work.
>
> > [1] Ozirmak Lermi, N., Molina Ayala, M., Hernandez, S. et al. Comparison of imaging based single-cell resolution spatial transcriptomics profiling platforms using formalin-fixed paraffin-embedded tumor samples. Nat Commun 16, 8499 (2025). https://doi.org/10.1038/s41467-025-63414-1
> >
> > [2] Lim H J, Wang Y, Buzdin A, et al. A practical guide for choosing an optimal spatial transcriptomics technology from seven major commercially available options[J]. BMC genomics, 2025, 26(1): 47.
>
> ---
>
> **5. Gene sentence construction:** We appreciate the reviewer’s point regarding gene selection strategy for pretraining. Due to strong batch effects across slides and platforms, **the pre-training stage requires a fixed and unified gene vocabulary**, rather than relying on slide-specific expression statistics. However, both HVG and SVG are computed directly from each individual slide’s expression or spatial pattern, and all spots within the same slide share the same selected gene set. This makes them inherently slide-level selections, **preventing them from providing a consistent semantic space at the spot level, and thus they cannot support a unified vocabulary for cross-slide pre-training.** In contrast, TF-IDF does not depend on a single slide’s statistics; it evaluates gene informativeness based on cross-sample frequency distribution, making it more suitable for pre-training.

---

> ### Author Response · Authors · 2025-11-25
>
> **6. Clarification on Equation (11):** We appreciate the reviewer’s comments on clarification. In our framework, model-predicted probabilities in Eq. 11 are produced by a projection head. After a histology patch is encoded into an embedding by the vision encoder, the projection head maps it into the same semantic space as the pathway prototypes. The mapped embedding then computes dot-product similarity with the prototype set embeddings, and the resulting scores are normalized using a softmax operation to yield the probability distribution $\hat{q}_i$. The soft target distribution $q_i(j) \propto \exp(\omega_j)$ is derived from the pathway–patch overlap, which injects structured biological prior knowledge. The KL divergence term in Eq. 11 encourages $\hat{q}_i$ to align with $q_i$, enabling biologically guided contrastive learning. A clarification has been added in Section 2.4 Multi-level Contrastive Alignment (Line 216).
>
> ---
>
> **7. Methodology and Motivation:** This work **does not only** aim to refine clustering task or spatial representation. **Instead, our goal is to address two key limitation of current ST foundation models, their lack of biological semantics and spatial organization in vision–omics representation modeling.** Therefore, we propose a representation alignment mechanism that integrates biological priors with region-level spatial structure to enable biologically interpretable alignment between histology and gene expression. This provides a principled representation basis for improving the pretraining of ST foundation models.
>
> ---
>
> **8. Parameter sizes:** We compare our model against representative digital pathology foundation models and summarize their parameter sizes and architectures in the Table 1 in revised manuscript. For vision–language datasets of approximately 1–2M paired samples, our design follows the same configuration as CONCH, with a ViT-Base visual encoder coupled with text transformer, which provides strong foundation-model capacity while maintaining computational efficiency.

---

> > ### Comment · Reviewer_hemv · 2025-11-27
> >
> > I appreciate the authors' detailed response. The rebuttal has clarified several aspects of the manuscript; however, significant concerns regarding the feasibility of high-resolution data and the empirical validation of the gene selection strategy remain unresolved. My remaining comments are as follows.
> >
> > **4. High-resolution datasets**
> >
> > I remain unconvinced by the exclusion of high-resolution datasets.
> >
> > Availability: The "limited scale" argument is invalid. Nicheformer (Nature Methods '25)[1] successfully trained on a massive dataset (SpatialCorpus-110M) with over 53 million spatially resolved cells from platforms like MERFISH and Xenium, proving that sufficient data exists for foundation models.
> >
> > Feasibility: The claim that distributional differences prevent integration is contradicted by HEST-1k (NeurIPS '24)[2]. This benchmark successfully pairs histology with multi-resolution data (Visium, Xenium, ST) in a unified framework, proving that technical integration is already a solved problem.
> >
> > Multi-resolution support is a current standard. I strongly recommend addressing this to validate generalizability.
> >
> > **5. Gene sentence construction**
> >
> > The theoretical preference for TF-IDF over HVG/SVG lacks empirical verification.
> >
> > Missing Baseline: The current ablation (Tables 4, 5, 7) only compares "HiBio-ST" against a version without TF-IDF. It fails to compare against the HVG/SVG strategies criticized in the rebuttal.
> >
> > Validation Required: To prove that TF-IDF is superior and that HVGs fail to support a unified vocabulary, a direct comparison (e.g., HiBio-ST with HVG selection) is necessary.
> >
> > Without experimentally showing that an HVG-based approach underperforms, the proposed method's validity remains unproven. I strongly suggest adding this comparison.
> >
> > ----
> > [1] Nicheformer: a foundation model for single-cell and spatial omics. Nature Methods (2025).
> >
> > [2] Hest-1k: A dataset for spatial transcriptomics and histology image analysis. NeurIPS (2024).

---

> > > ### Author Response · Authors · 2025-11-30
> > >
> > > Thank you for recognizing our efforts. Here, we would like to address your remaining concerns and questions:
> > >
> > > ---
> > > **1. Performance on High-Resolution Dataset:**
> > >  We fully understand the reviewer’s concern of validation on multi-resolution spatial transcriptomics data. In response, we conducted additional experiments on two high-resolution (Visium HD) datasets under two different spatial resolution settings (8 $\mu$m and 16 $\mu$m) to assess the stability of our model across multiple spatial scales. The corresponding results are summarized in Table 8 (Line 1242) and Table 9 (Line 1296) of the revised manuscript, with detailed analysis provided in Supplementary Section *Performance on High-Resolution Dataset* (Line 1325). Across both datasets, both resolutions, and multiple downstream tasks, our method consistently achieves the best performance on the majority of evaluation metrics, demonstrating that the proposed model remains stable and highly effective in multi-resolution spatial transcriptomics scenarios.
> > >
> > > > We further selected two high-resolution Visium HD datasets to evaluate the model's performance under high-resolution spatial transcriptomics conditions, including the Breast Cancer HD dataset (3 slides) [1] and the CRC HD dataset (5 slides) [2]. To ensure consistency with our original visual input design, we cropped 224×224 px image patches centered at each bin, and conducted experiments under two resolution settings: 8 $\mu$m and 16 $\mu$m. In terms of experimental protocol, we used 3-fold cross-validation for the Breast Cancer HD dataset and 5-fold cross-validation for the CRC dataset. In terms of experimental protocol, we used 3-fold cross-validation for the Breast Cancer HD dataset and 5-fold cross-validation for the CRC dataset. Table 8 and Table 9 present the results of various methods on these high-resolution datasets for the gene expression prediction task and the image-to-ST-sentence retrieval task, respectively, demonstrating that the proposed model maintains stable and superior effectiveness under multi-resolution settings.
> > > >
> > > > [1] Nagendran M, Sapida J, Arthur J, et al. Visium HD enables spatial discovery in FFPE human breast cancer at single-cell scale[EB/OL].(2023)
> > > >
> > > > [2] Oliveira M F, Romero J P, Chung M, et al. High-definition spatial transcriptomic profiling of immune cell populations in colorectal cancer[J]. Nature Genetics, 2025: 1-12.
> > >
> > > It is worth noting that, unfortunately, during the period in which this study was conducted (by 09/2025), the publicly available integrated dataset HEST-1K or STimage-1K4M, although providing cross-platform high-resolution slides, contained only 75 Xenium/HD slides, representing merely ~6.1% of the dataset, while Visium + STv1 accounted for 1,154 slides. Given this highly imbalanced data distribution, we chose to train our model only on the mainstream platforms (Visium/ST) to ensure stable and reliable learning.
> > >
> > > We sincerely appreciate the efforts including HEST-1K, STimage-1K4M, and Nicheformer teams for publicly consolidating multi-platform datasets, as well as the reviewer’s constructive suggestion. As more high-resolution spatial omics data become more available, we will continue expanding our base vocabulary and pretraining strategy, and aim to incorporate joint cross-resolution pretraining in future versions of our work.

---

> > > ### Author Response · Authors · 2025-11-30
> > >
> > > **2. Comparison with HVG selection:**
> > >  We appreciate the reviewer’s constructive suggestion. Following the advice, we included an additional baseline that constructs gene sentences using the High Variance Gene (HVG) selection strategy under the same experimental settings. Specifically, we computed gene variance at the slide level and selected high-variance gene names within each slide to form a shared HVG gene sentence, replacing the TF-IDF–based spot-specific selection strategy. This HVG-based approach follows exactly the same pretraining and downstream evaluation protocols as our original method. The corresponding results have been added to Table 4, Table 5, Table 7, and Figure 7, with detailed analysis provided in Supplementary Section *Comparison with HVG Gene Selection* (Line 1294).
> > >
> > > > To evaluate the effectiveness of our gene selection strategy, we conducted additional experiments using the High Variance Gene (HVG) selection method under the same experimental settings. Specifically, we computed gene variance at the slide level and selected high-variance genes within each slide to construct a shared HVG gene sentence, meaning that all spots from the same slide use an identical HVG vocabulary. This HVG-based strategy follows exactly the same pretraining and downstream evaluation procedures as our original method. The corresponding results have been added to Table 4, Table 5, Table 7, and Figure 7.
> > > >
> > > > The experimental results show that the model exhibits a clear performance drop on almost all downstream tasks under this setting, with most results even falling below those obtained using the Top-gene selection strategy. We attribute this phenomenon to the fact that HVG selection relies on slide-level variance statistics, thus providing only bulk-level shared gene information and failing to capture the fine-grained, spot-level biological semantics. In contrast, both TF-IDF and Top-gene strategies operate on the expression distribution of individual spots, thereby preserving spatial heterogeneity and localized distribution patterns. As a result, these two spot-specific strategies demonstrate greater stability and superiority in spot-level cross-modal tasks. Thus, we validate that TF-IDF is a more suitable and effective strategy for constructing gene sentences and establishing a unified semantic vocabulary for pretraining.

---

### Official Review · Reviewer_RMdn · 2025-11-01

**Soundness:** 2
**Presentation:** 2
**Contribution:** 2
**Rating:** 4
**Confidence:** 4

**Summary:**

This paper introduces the HiBio-ST pretraining framework, designed to integrate histology images and gene expression for spatial transcriptomics analysis. The model first employs a TF-IDF reweighting scheme to highlight spatially informative genes. It then incorporates curated pathways to inject biological priors and guide the alignment. Finally, a hierarchical, region-aware clustering method groups spots into coherent meso-scale structures, allowing the model to capture higher-order spatial patterns. The authors evaluate their model on multiple downstream tasks and achieve better performance than the baseline method.

**Strengths:**

1. The idea of using the TF-IDF reweighting to find important genes and integrating pathway anchors to inject biological priors is novel.
2. The authors evaluate their method on several datasets, including gene expression prediction, zero-shot layer clustering, and image-to-sentence retrieval, with good performance.

**Weaknesses:**

1. The evaluation is limited to a few datasets/organs. Why not evaluate the HEST benchmark to show the model’s generalizability? Consequently, it’s insufficient validation for a called "foundation model".
2. The paper lacks an analysis of important hyperparameters
3. The choice of K in the TF-IDF reweighting scheme and its impact on performance.
- The threshold used for the pathway-guided alignment, and how different values (higher vs. lower) would affect the results.
- The paper would be significantly strengthened by providing visualizations that illustrate how the region identifiers evolve during the training process.
4. The authors should provide an ablation study on the weights for different loss component to show their contribution to the final performance.

**Minor**:
1. The reported performance of some baselines (such as TRIPLEX) appears unusually low. The authors should verify their implementation or justify the result.
2. The paper lacks justification for the chosen image encoder. It is unclear why more recent, domain-specific vision foundation models (e.g., UNI or Virchow) were not considered, as they might offer stronger performance.
3. The authors should include a comparison against more recent, relevant models (such as CONCH [1]) as a baseline for the image-to-ST sentence retrieval task.
4. Clarification for Eq. 11: The equation computes a KL divergence using "the model's predicted probabilities," but it is not apparent which component of the model (as shown in Figure 1) generates these probabilities. The authors should clarify this part.

[1] Lu, Ming Y., et al. "A visual-language foundation model for computational pathology." Nature Medicine 30.3 (2024): 863-874.

**Questions:**

Please see the Weaknesses section.

---

> ### Author Response · Authors · 2025-11-25
>
> **1. Downstream Datasets:** We understand the reviewer’s concern about the evaluation scope. Actually, HEST-1K dataset is currently the most comprehensive ST collection available, containing over 150 cohorts, 1,000 slides, and 2 million spatial spots. **In our work, we use the majority of HEST-1K (949 slides and ~1.5M spots from 149 cohorts, covering 26 organs across two species) as the pretraining corpus.** **To prevent data leakage, we exclude four widely-used ST datasets with 3 different organs** from the pretraining pool and reserve them solely for downstream evaluation.
>
> **Given current limited availability** of large-scale, independent ST datasets in the field, it is challenging to construct a fully non-overlapping benchmark beyond HEST-1K. While our evaluation already spans multiple organs, multiple platforms, and several clinically distinct use cases, providing a comprehensive assessment of generalizability under the constraints of current ST data resources.
>
> ---
>
> **2. Hyperparameter Sensitivity Analysis:** We appreciate the reviewer’s suggestion. Following the suggestion, we incorporated hyperparameter sensitivity analyses, with additional results reported in Tables 4, 5, and 7, and corresponding discussions added in *Section Pathway Overlap Ratio* (Line 481) and *Section Loss Weights Analysis (Line 490)*.
>
> > **Pathway Overlap Ratio.**  We evaluate two configurations ($\tau = 0.1$ and $\tau = 0.5$) in comparison with our default setting ($\tau = 0.2$) to better understand the influence of hyperparameters in our framework with results summarized in Tables 4, 5, and 7. The results indicate that a low overlap ratio leads to noticeable performance degradation. This occurs because reducing $\tau$ introduces an excessive number of positive pathway pairs, thereby increasing the probability of incorporating noisy or biologically irrelevant pathways as anchors. In contrast, when $\tau$ becomes overly strict, the performance either plateaus or slightly decreases, as the number of valid pathway anchors becomes too limited to provide sufficient biological supervision during contrastive alignment. Thus, a moderate overlap ratio yields the best trade-off between eliminating noisy anchors and retaining informative and biologically meaningful pathway anchors.
> >
> > **Loss Weights Analysis.** We evaluate different configurations of the loss weights $\lambda_2$ and $\lambda_3$ in our hybrid optimization objective, with results summarized in Tables 4, 5 and Table 7. We observe that increasing both loss weights consistently improves performance across multiple downstream tasks, indicating that stronger supervision from biological pathway anchors provides more informative and structured guidance to the representation space. In other words, enhancing the contribution of biological priors encourages the model to more effectively align visual features with functional semantics, ultimately strengthening biologically meaningful learning. However, we also find that excessively large loss weights introduce instability during training, where gradients fluctuate and convergence becomes less reliable. This suggests that while biological priors are beneficial, overly dominating them may interfere with the balance among different learning objectives.
> ---
> **3. Token Length of Gene Sentence:** We conducted an ablation study to analyze how different choices of K in our TF-IDF–based gene selection influence model performance. The results are included in Table 4, where we compare multiple values of K across downstream datasets. In addition, we provide a more detailed explanation of this phenomenon in Section Token Length of Gene Sentence (Line 514).
>
> > **Token Length of Gene Sentence.** We further investigate the impact of gene sentence length on model performance by varying the token length from 100 to 500. As shown in Table 4, using a longer gene sentence (250 tokens) consistently leads to better performance across all datasets and metrics, suggesting that incorporating more gene-level information helps the model learn more comprehensive representations for downstream prediction tasks. However, increasing the token length beyond 250 introduces slight performance degradation in some settings (e.g., Breast Cancer MSE increases from 0.6082 to 0.6406), likely due to informational noise or redundancy introduced by less informative or lowly expressed genes. These results suggest a trade-off: while longer sentences enrich the biological context, excessively long sequences may overwhelm the model with irrelevant information or exceed its capacity to focus on key genes. Thus, selecting an optimal token length is crucial for balancing expressiveness and robustness in multimodal learning.

---

> ### Author Response · Authors · 2025-11-25
>
> **4. Threshold of Pathway Overlap Ratio:** We appreciate the reviewer’s suggestion regarding the threshold used for the pathway–patch alignment ratio. Following the reviewer’s suggestion, we conducted supplementary experiments using $\tau = 0.1$ and $\tau = 0.5$ in addition to our default $\tau = 0.2$. The new results are presented in Tables 4, 5, and 7, with a detailed discussion added in Section Pathway Overlap Ratio (Line 481).
>
> >  We evaluate two configurations ($\tau = 0.1$ and $\tau = 0.5$) in comparison with our default setting ($\tau = 0.2$) to better understand the influence of hyperparameters in our framework with results summarized in Tables 4, 5, and 7. The results indicate that a low overlap ratio leads to noticeable performance degradation. This occurs because reducing $\tau$ introduces an excessive number of positive pathway pairs, thereby increasing the probability of incorporating noisy or biologically irrelevant pathways as anchors. In contrast, when $\tau$ becomes overly strict, the performance either plateaus or slightly decreases, as the number of valid pathway anchors becomes too limited to provide sufficient biological supervision during contrastive alignment. Thus, a moderate overlap ratio yields the best trade-off between eliminating noisy anchors and retaining informative and biologically meaningful pathway anchors.
> ---
> **5. Region Identifiers Update:** We appreciate the reviewer’s insightful suggestion regarding the visualization of region identifiers. To address this point, we conducted additional pretraining experiments with HER2 samples that contain expert‐annotated spatial clusters (6 groups). We then visualized how the learned region identifiers progressively evolve throughout the training process. The corresponding results are presented in Figure 9, and a detailed discussion is provided in Supplementary Section D.1: Region Identifier Updates (Line 1126).
>
> > **Region Identifier Updates.** We conducted additional pretraining experiments on HER2 samples containing expert‐annotated spatial clusters (6 groups) to visualize the progressive refinement of regional identity alignment for each spot. As shown in Figure 9, at the initial stage, all spots are initialized to a single group, reflecting the absence of spatially meaningful structure. After several updates, the number of region groups gradually decreases, and adjacent spots begin to aggregate into coherent domains. This refinement emerges from the combined signals of ST gene similarity and visual feature proximity, which guide nearby spots toward consistent regional identities. During this process, the spatial layout becomes more structured and less chaotic, forming biologically interpretable tissue regions.
> ---
> **6. Weights of loss function:** We appreciate the reviewer’s suggestion regarding the weights for loss component. Following the suggestion, we conducted a sensitivity analysis on these loss weights. Specifically, we evaluated two additional configurations (weights $\lambda_2$ = 0.25 $\lambda_3$ = 0.1 and $\lambda_2$ = 0.5 $\lambda_3$ = 0.5 ) alongside our default setting ($\lambda_2$ = 0.25 $\lambda_3$ = 0.25 ). The new results are presented in Tables 4, 5, and 7, with a detailed discussion added in Section Loss Weights Analysis (Line 490).
>
> > **Loss Weights Analysis.** We evaluate different configurations of the loss weights $\lambda_2$ and $\lambda_3$ in our hybrid optimization objective, with results summarized in Tables 4, 5 and Table 7. We observe that increasing both loss weights consistently improves performance across multiple downstream tasks, indicating that stronger supervision from biological pathway anchors provides more informative and structured guidance to the representation space. In other words, enhancing the contribution of biological priors encourages the model to more effectively align visual features with functional semantics, ultimately strengthening biologically meaningful learning. However, we also find that excessively large loss weights introduce instability during training, where gradients fluctuate and convergence becomes less reliable. This suggests that while biological priors are beneficial, overly dominating them may interfere with the balance among different learning objectives.

---

> ### Author Response · Authors · 2025-11-25
>
> **7. Performance of TRIPLEX:** The lower performance of TRIPLEX in our setting can be attributed to two key differences from its original configuration:
>
> (1)  **Gene expression normalization.** The original TRIPLEX paper adopts library-size log normalization, whereas our standardized setup applies log1p normalization across all methods. Differences in preprocessing pipelines are known to affect ST modeling performance and likely contribute to the performance gap.
>
> (2)  **Data split strategy.** We use sample-level cross-validation, where training and testing slides come from different samples. TRIPLEX was originally evaluated under different data distributions, and cross-sample variation can significantly impact model generalization.
>
> Despite these differences, we confirm that all baselines, including TRIPLEX, were trained and evaluated under exactly the same experimental protocol, ensuring a fair comparison.
>
> ---
>
> **8. Modality-specific Foundation Models:** We acknowledge that modality-specific foundation models offer strong within-modality representations. Although domain-specific vision foundation models such as UNI and Virchow provide strong representations within a single modality, their pretraining objectives (e.g., patch contrastive learning) are not designed for cross-modal alignment between histology images and spatial transcriptomics. As a result, the thorough training process drives the model toward a new cross-modal optimum rather than preserving the pretrained structure. To further support our clarification, we include zero-shot clustering and few-shot expression prediction results for UMPIRE [1] and STPath[2], both of which adopt pretrained CONCH and GigaPath as their vision encoders. The performance comparison is provided in Figure 2 and Table 1.
>
> > [1] Han M, Yang D, Cheng J, et al. Towards unified molecule-enhanced pathology image representation learning via integrating spatial transcriptomics[J]. Pattern Recognition, 2025: 112458.
> >
> > [2] Huang T, Liu T, Babadi M, et al. STPath: a generative foundation model for integrating spatial transcriptomics and whole-slide images[J]. npj Digital Medicine, 2025, 8(1): 659.
> ---
> **9. Baseline Performance in image-to-ST sentence retrieval task:**
> We thank the reviewer for the helpful suggestion. Following recommendation, we have added CONCH as an additional baseline for the image-to-ST sentence retrieval task. The new results are included in Table 3 the revised manuscript and show that our method consistently outperforms CONCH across all datasets and metrics.
>
> ---
> **10. KL divergence:** We appreciate the reviewer’s comments on clarification. In our framework, model-predicted probabilities in Eq. 11 are produced by a projection head. After a histology patch is encoded into an embedding by the vision encoder, the projection head maps it into the same semantic space as the pathway prototypes. The mapped embedding then computes dot-product similarity with the prototype set, and the resulting scores are normalized using a softmax operation to yield the probability distribution $\hat{q}_i$. The soft target distribution $q_i(j) \propto \exp(\omega_j)$ is derived from the pathway–patch overlap, which injects structured biological prior knowledge. The KL divergence term in Eq. 11 encourages $\hat{q}_i$ to align with $q_i$, enabling biologically guided contrastive learning. A clarification has been added in Section 2.4 Multi-level Contrastive Alignment (Line 216)

---

### Official Review · Reviewer_xsAL · 2025-11-01

**Soundness:** 3
**Presentation:** 4
**Contribution:** 3
**Rating:** 6
**Confidence:** 4

**Summary:**

This paper introduces HiBio-ST, a new pretraining method for spatial transcriptomics foundation model. Instead of normally used SVG, HVG embedding/processing of gene table, each site read out is first ranked by TF-IDF algorithm on the gene sections, then chosen the top-k most genes from this ranking as a gene sentence. After this set of sentences is constructed, these sentences are being cross-referenced with the ‘pathway’ which are known biological functions of sets of genes (KEGG pathways is what the authors chose). On the imaging side, instead of just learning the site level embeddings independently, the authors try to incorporate regions that are essentially clusters generated from spatial locations and image-gene similarity. The dataset

**Strengths:**

1. This paper is well-written. It is clear, easy to follow, and well-motivated.
2. This paper tries to improve the foundation model pre-training specifically on the ST dataset. It introduces several novel points. First it treats the gene as sentences and uses the NLP method to conduct initial dimension reduction. Secondly, it has a biology reference bit where it incorporates biological knowledge as priors which prior works lack. Finally, it improves the imaging side to allow spots to have regions or a spatial sense rather than being treated independently.
3. The reviewer really appreciates the clear and logical flow of the ablations study or the flow of paper writing in general. While reading the methods sections, the reviewer writes down a few questions on the different design bits, but they are mostly answered or at least experimented in the ablation section.

**Weaknesses:**

1. The reviewer thinks Figure 3 should be in appendix as it is a zero-shot layer clustering task where the qualitative results are all pretty bad or different from the human annotation. Putting it here takes a lot of space but not telling too much information.
2. The figures shown in the paper are in general too small, at least in an arms length distance.
3. If treating gene readout as keywords or sentences, there should be some other way to conduct such dimension reduction or ranking or selection. This work does not consider or discuss those possibilities.
4. Clustering into regions does provide spatial correlations between spots but there is a great amount of work incorporating graphs for spatial correlations, this work does not mention why or why not those were not being considered.

**Questions:**

1. For the four tasks, since the reviewer does not come from a computational biology background, it would be hard for the reviewer to assess even excelling in these tasks. How useful would a ST foundation model be applied if given to all the cancer researchers?
2. Is the KEGG pathway the only or complete definition of genes? Is the field of gene pathways still involved and at what speed? Do the KEGG pathways be useful or correlated for the gene set in the dataset? If Visum changed their generation or readout protocols would these prior still hold? Will biological knowledge shift or change? (all the above is quite similar question, the reviewer thinks that the authors should illustrate more on the importance or stability of such priors)
3. Is the choice of top-K would be similar to the experiment for the 100-500 tokens?
4. How does the choice of ViTs and Text encoders or image/gene encoders in general affect the performance or pretraining? Would foundation models from each modality (pathology and RNA) help here?


The reviewer is holding a positive view on this paper as it flows nicely and the ablation study is well-done. The reviewer is giving a borderline acceptance and willing to change scores during the discussion session.

---

> ### Author Response · Authors · 2025-11-25
>
> **1. Purpose and biological interpretation for downstream tasks:** We appreciate the reviewer’s comment on the purpose and interpretation of the downstream benchmarking tasks. Our work aims to **leverage easily accessible histology images to analyze spatial transcriptomics (ST), thereby reducing the dependency on costly ST profiling.** To comprehensively **evaluate the biological utility of our histology-derived representations**, we design a series of downstream benchmarking tasks that reflect distinct biological objectives. We clarify the specific purpose and biological interpretation of each task in Supplementary Section C.2 Implementation Details for Downstream Tasks (Line 966).
>
> > **(1) Zero-shot Spatial Clustering**
> >
> > *Purpose:* This task demonstrates the model's ability to produce biologically meaningful spatial structures directly from histology-derived embeddings without any supervision [1]. It examines whether the pretrained model can implicitly generate and organize spatial representations that correspond to real tissue architectures.
> >
> > *Biological Interpretation:* A model capable of zero-shot clustering that aligns with known tissue architectures indicates that its histology-based representations preserve biologically meaningful spatial patterns, reflecting the ability to capture inherent tissue heterogeneity purely from visual cues.
> >
> > **(2) Few-shot Prediction & Gene Expression Prediction**
> >
> > *Purpose:* Few-shot prediction assesses the generalization ability of a foundation model under limited ST supervision or with only minimal annotated data. Gene expression prediction further evaluates the model’s ability to infer ST profiles directly from histology images, examining how effectively visual morphology can recover underlying molecular signals [2].
> >
> > *Biological Interpretation:* These tasks reflect practical scenarios where only a small portion of the tissue is profiled with ST. This demonstrates how histology can serve as a surrogate to recover biologically meaningful molecular signals when ST data is sparse or unavailable.
> >
> > **(3) Image-to-ST Sentence Retrieval**
> >
> > *Purpose:* This task evaluates the model’s ability to align and maintain consistency between tissue morphology and higher-level biological functions [3].
> >
> > *Biological Interpretation:* Each gene set represents a distinct biological process or phenotype. Successful retrieval indicates that the model effectively learns biologically interpretable visual–molecular relationships and can reliably associate histological regions with their relevant functional gene sets.
> >
> > [1] Maynard K R, Collado-Torres L, Weber L M, et al. Transcriptome-scale spatial gene expression in the human dorsolateral prefrontal cortex[J]. Nature neuroscience, 2021, 24(3): 425-436.
> >
> > [2] He B, Bergenstråhle L, Stenbeck L, et al. Integrating spatial gene expression and breast tumour morphology via deep learning[J]. Nature biomedical engineering, 2020, 4(8): 827-834.
> >
> > [3] Chen W, Zhang P, Tran T N, et al. A visual–omics foundation model to bridge histopathology with spatial transcriptomics[J]. Nature Methods, 2025: 1-15.

---

> ### Author Response · Authors · 2025-11-25
>
> **2. Pathway Selection:** We appreciate the reviewer’s comments regarding the importance and stability of the biological priors. We provide a consolidated response as follows.
>
> > (1) We agree that KEGG is not the only pathway database. Other pathway systems such as Reactome, MetaCyc, and MSigDB are also widely used in transcriptomic analyses. Therefore, **we do not treat KEGG as the “only” or “complete” definition of gene relationships;** rather, we adopt it as a well‐established, structurally stable, and broadly validated biological prior that has been extensively used in spatial omics and cancer studies.
> >
> > (2) The core metabolic and signaling networks included in KEGG **have remained highly stable over the past decade**, with updates mainly involving incremental extensions rather than structural rewrites. These steady-state biological structures are widely adopted across transcriptomic and spatial transcriptomic studies [1-3].
> >
> > (3)  The KEGG structural prior is also well aligned with the gene set in our dataset. In our intersection analysis, we observed substantial overlap between Visium-captured genes and core KEGG pathway genes, particularly those related to tumor microenvironment remodeling, etc. [4,5].
> >
> > (4)  Pathway priors represent intrinsic biological relationships among genes rather than platform-specific technical noise. While changes in Visium protocols may alter capture efficiency or sequencing depth, they do not affect fundamental gene–gene interactions within canonical pathways. Thus, KEGG pathways function as platform-independent biological priors that remain valid across protocol versions.
> >
> > (5)  Finally, we acknowledge that biological knowledge continues to evolve. To ensure long-term adaptability, our framework is designed to allow the pathway database to be replaced. **The model does not rely on a specific version of KEGG**, but instead operates on the abstract concept of structured pathway organization. Future updates to KEGG, Reactome, or other pathway resources can be incorporated simply by updating the gene–pathway mappings, without requiring any changes to the model architecture, ensuring natural compatibility with evolving biological knowledge.
> >
> > [1] Kanehisa M, Goto S, Sato Y, et al. KEGG for integration and interpretation of large-scale molecular data sets[J]. Nucleic acids research, 2012, 40(D1): D109-D114.
> >
> > [2] Kanehisa M. Toward understanding the origin and evolution of cellular organisms[J]. Protein science, 2019, 28(11): 1947-1951.
> >
> > [3] Kanehisa M, Furumichi M, Sato Y, et al. KEGG: integrating viruses and cellular organisms[J]. Nucleic acids research, 2021, 49(D1): D545-D551.
> >
> > [4] Liu Z, Zhang Z, Zhang Y, et al. Spatial transcriptomics reveals that metabolic characteristics define the tumor immunosuppression microenvironment via iCAF transformation in oral squamous cell carcinoma[J]. International Journal of Oral Science, 2024, 16(1): 9.
> >
> > [5] Hakobyan S, Schmidt M, Binder H, et al. Topology-aware pathway analysis of spatial transcriptomics[J]. PeerJ, 2025, 13: e19729.
>
> **3. Top-K setting:** We appreciate the reviewer’s comments regarding top-K selection setting. The experimental trends observed in the paper can indeed be viewed as analogous to the performance changes under different top-gene selection strategies. Specifically, we compared OmiCLIP (76 tokens, corresponding to ~20–30 genes) with our w.o TF-IDF 250-token (250 genes) ablation setting. Although the two methods differ in architecture and tokenization (OmiCLIP uses ViT-Large with BPE, whereas our method uses ViT-Base with gene-specific tokens), their performance trends can still be meaningfully compared.  The results show that increasing the token count covers more genes but also introduces additional noise and redundancy. As the top-gene set grows, high-frequency or convergent gene tokens dilute the embedding’s discriminative power, leading to decline in retrieval accuracy and gene reconstruction (Table 2&4, Table 3&5).

---

> ### Author Response · Authors · 2025-11-25
>
> **4. Encoder Selection:** We appreciate the reviewer’s comment.
>
> **(1)**  **Encoder Selection:** We believe that model capacity should be aligned with the available data scale. We summarize in Table (Line 348) the vision encoders, parameter scales, and clustering performance of mainstream pathology foundation models. Prior studies in digital pathology have shown that with approximately 1–2M image patches, ViT-Base is already sufficient to learn rich morphological features [1], whereas scaling up to ViT-L/H does not ensure additional gains and could lead to overfitting or unstable training.
>
> Enlarging the text/gene encoder is even more costly, as the large number of gene tokens significantly increases the computational burden of self-attention. Therefore, under our current data scale and task setting, using a ViT-Base backbone together with a lightweight text/gene encoder represents a stable capacity–efficiency trade-off. At the same time, we acknowledge that with larger datasets or more advanced architectures, GPT-style or CoCa-style encoders may potentially provide further improvements, which is an interesting direction for future exploration.
>
> **(2)**  **Modality-specific Foundation Models:** We acknowledge that modality-specific foundation models offer strong within-modality representations. **However, their pretraining objectives differ fundamentally from our task, which focuses on cross-modal alignment between histology images and spatial transcriptomics.** Pathology foundation models are typically trained with patch-level image modeling, while RNA foundation models (e.g., scGPT) emphasize gene-sequence modeling. These objectives do not naturally align the two modalities within a shared embedding space. As a result, the thorough training process drives the model toward a new cross-modal optimum rather than preserving the pretrained structure. At the same time, we agree that with unified multimodal pretraining frameworks, foundation models may offer additional benefits and represent a promising future direction.
>
>  To further support our clarification, we include zero-shot clustering and few-shot expression prediction results for UMPIRE [2] and STPath[3], both of which adopt pretrained CONCH and GigaPath on HEST-1K as their vision encoders. The performance comparison is provided in Figure 2 and Table 1.
>
> > [1] Lu M Y, Chen B, Williamson D F K, et al. A visual-language foundation model for computational pathology[J]. Nature medicine, 2024, 30(3): 863-874.
> >
> > [2] Han M, Yang D, Cheng J, et al. Towards unified molecule-enhanced pathology image representation learning via integrating spatial transcriptomics[J]. Pattern Recognition, 2025: 112458.
> >
> > [3] Huang T, Liu T, Babadi M, et al. STPath: a generative foundation model for integrating spatial transcriptomics and whole-slide images[J]. npj Digital Medicine, 2025, 8(1): 659.
>
> **5.Gene sentence Construction:** We appreciate the reviewer’s comments .
> 1. In our design, the gene names, rather than the expression readouts, are treated as “words/tokens’’ in the sentence construction. The expression readouts are only used to rank and select which gene tokens should appear. Therefore, our TF-IDF strategy is not a dimensionality reduction of expression values, but a biologically informed token selection mechanism.
> 2. Moreover, due to substantial batch effects in ST data across cohorts and sequencing platforms, raw readouts can vary dramatically and are difficult to be directly used as a stable basis for token construction. In contrast, selecting gene names via TF-IDF yields a more platform-robust vocabulary.
> 3. Existing ranking-only strategies (e.g., GenePT [1], OmiCLIP), which construct sentences by simply ordering genes by expression levels, are often dominated by highly abundant housekeeping genes and may underrepresent low-to-moderate but spatially informative signals. Our pathway-guided TF-IDF selection mitigates these issues by down-weighting ubiquitous genes and emphasizing biologically relevant spatial markers.
>
> > [1] Chen Y, Zou J. GenePT: a simple but effective foundation model for genes and cells built from ChatGPT[J]. bioRxiv, 2024: 2023.10. 16.562533.
>
> **6. Spatial Correlation:** We appreciate the reviewer’s insightful suggestion. In this work, our focus is to leverage biological priors and region-level spatial information to improve the alignment between histology and omics representations, rather than to just design a new spatial aggregation method. Graph-based spatial modeling is not contradictory to our approach and may serve as a complementary strategy for capturing more fine-grained spatial interactions. We appreciate this helpful suggestion and will include it as a promising future direction.
>
> **7. Figure Size:** We appreciate the suggestion on readability. Due to page limits, we scaled down several figures for layout constraints. In the revised version, we have enlarged these figures to improve readability and visual clarity.

---

> ### Comment · Reviewer_xsAL · 2025-11-26
> **ACK**
>
> Thanks for all the detailed reply and revision. I maintain my initial rating and do think this paper is well-written and have its own contirbution to the field.

---

> > ### Author Response · Authors · 2025-11-26
> >
> > Thank you for your positive feedback and for acknowledging the improvements in our manuscript. We sincerely appreciate your constructive review and valuable suggestions.

---

### Official Review · Reviewer_d1zK · 2025-11-11

**Soundness:** 2
**Presentation:** 3
**Contribution:** 2
**Rating:** 4
**Confidence:** 4

**Summary:**

This study presents a novel visual-omics foundation model, HiBio-ST, that integrates spatial and biological pathway information through a TF-IDF reweighting scheme and hierarchical modeling. The framework highlights spatially informative genes, incorporates pathway-level priors, and introduces a region-aware contrastive alignment mechanism. The model is evaluated on several downstream tasks, including clustering, and gene expression prediction.

**Strengths:**

1. This paper lies in its unique incorporation of spatial and biological pathway information into a visual-omics foundation model, effectively bridging morphology and molecular representation learning. A key innovation over existing approaches is the TF-IDF reweighting strategy, which selects genes that are locally enriched (captured by the TF term) but not globally expressed (filtering out housekeeping genes), a biologically meaningful and effective approach.

2. The paper also presents a series of ablation studies that demonstrate the contribution of each model component to the overall framework.

**Weaknesses:**

Despite the methodological novelty, the experimental validation leaves several important aspects unconvincing. In particular, the spatial clustering results on the DLPFC dataset are not strong, the predicted domains fail to recover clear laminar structures and appear inconsistent with the known cortical anatomy. This raises questions about whether the proposed biological priors truly enhance spatial representation quality. The manuscript would benefit from additional sanity checks showing that the embeddings preserve biologically meaningful relationships (e.g., spots from the same cortical layer or with similar cell-type compositions clustering together).

Furthermore, while the benchmarking includes comparisons with other foundation models, it omits direct comparisons with task-specific spatial analysis tools such as GraphST or STAGATE, which are designed for spatial domain detection. Without these baselines, it is difficult to assess whether HiBio-ST offers practical improvements for biologically relevant tasks.

Finally, although the method is systematically evaluated across multiple tasks, the presentation tends to emphasize quantitative improvements without sufficient biological interpretation or qualitative analysis, making it difficult to judge whether the gains are meaningful from a biological perspective.

**Questions:**

1. Could the authors clarify, in the ablation study under w/o TF-IDF, whether the same number of genes (250) are used but selected based on top-expressed genes? This clarification is important to demonstrate the advantage of the TF-IDF strategy over the commonly used selection approach.
2. Could the authors perform a sensitivity analysis on the threshold used for the indicator of the pathway-patch alignment score?
3. Could the authors explain the specific purpose and biological interpretation of each downstream benchmarking analysis?
4. Could the authors clarify the transformations applied (e.g., log(1 + x)) for each benchmarking method and state whether the implementations are consistent and fair?
5. Could the authors repeat the experiments with different random seeds to evaluate the stability of the results under randomness?
6. Could the authors report the results of spatial clustering without the spatial label-smoothing mechanism, as HiBio-ST theoretically already captures spatial information and should outperform other methods without redundant spatial regularization?
7. It is unclear whether the authors considered simpler baselines or alignment strategies, such as using a single-scale InfoNCE loss without hierarchical extensions. The proposed framework appears to stack multiple existing components (TF-IDF weighting, pathway priors, spatial clustering, and multi-level contrastive learning) without a clear unifying principle or demonstrated necessity. The ablation studies only show marginal differences and do not isolate whether the hierarchical contrastive design truly contributes beyond increased model complexity. Could the authors justify why this multi-stage architecture is required, and provide evidence that its improvements are not merely due to overparameterization or better hyperparameter tuning?

---

> ### Author Response · Authors · 2025-11-25
>
> **1. Experiment setting for ablation study:** We appreciate the reviewer’s constructive comment on the ablation study setup. In the w/o TF-IDF setting, we indeed keep the same number of genes (250) to ensure a fair comparison. These genes are selected based on the top-expressed gene ranking, which is a commonly used strategy in existing spatial transcriptomics pipelines. This clarification has been explicitly added in Section Biological Information Prior (Line 463) of the revised manuscript and highlighted in blue.
>
> **2. Threshold of Pathway Overlap Ratio:** We appreciate the reviewer’s suggestion regarding the threshold used for the pathway–patch alignment ratio. Following the reviewer’s suggestion, we conducted supplementary experiments using $\tau = 0.1$ and $\tau = 0.5$ in addition to our default $\tau = 0.2$. The new results are presented in Tables 4, 5, and 7, with a detailed discussion added in Section Pathway Overlap Ratio (Line 481).
>
> >  We evaluate two configurations ($\tau = 0.1$ and $\tau = 0.5$) in comparison with our default setting ($\tau = 0.2$) to better understand the influence of hyperparameters in our framework with results summarized in Tables 4, 5, and 7. The results indicate that a low overlap ratio leads to noticeable performance degradation. This occurs because reducing $\tau$ introduces an excessive number of positive pathway pairs, thereby increasing the probability of incorporating noisy or biologically irrelevant pathways as anchors. In contrast, when $\tau$ becomes overly strict, the performance either plateaus or slightly decreases, as the number of valid pathway anchors becomes too limited to provide sufficient biological supervision during contrastive alignment. Thus, a moderate overlap ratio yields the best trade-off between eliminating noisy anchors and retaining informative and biologically meaningful pathway anchors.
>
>
> **3. Purpose and biological interpretation for downstream tasks:** We appreciate the reviewer’s comment on the purpose and interpretation. Our work aims to **leverage easily accessible histology images to analyze spatial transcriptomics (ST), thereby reducing the dependency on costly ST profiling.** To comprehensively **evaluate the biological utility of our histology-derived representations**, we design a series of downstream benchmarking tasks that reflect distinct biological objectives. We clarify the specific purpose and biological interpretation of each task in Supplementary Section C.2 Implementation Details for Downstream Tasks (Line 966).
>
> > **(1) Zero-shot Spatial Clustering**
> >
> > *Purpose:* This task demonstrates the model's ability to produce biologically meaningful spatial structures directly from histology-derived embeddings without any supervision [1]. It examines whether the pretrained model can implicitly generate and organize spatial representations that correspond to real tissue architectures.
> >
> > *Biological Interpretation:* A model capable of zero-shot clustering that aligns with known tissue architectures indicates that its histology-based representations preserve biologically meaningful spatial patterns, reflecting the ability to capture inherent tissue heterogeneity purely from visual cues.
> >
> > **(2) Few-shot Prediction & Gene Expression Prediction**
> >
> > *Purpose:* Few-shot prediction assesses the generalization ability of a foundation model under limited ST supervision or with only minimal annotated data. Gene expression prediction further evaluates the model’s ability to infer ST profiles directly from histology images, examining how effectively visual morphology can recover underlying molecular signals [2].
> >
> > *Biological Interpretation:* These tasks reflect practical scenarios where only a small portion of the tissue is profiled with ST. This demonstrates how histology can serve as a surrogate to recover biologically meaningful molecular signals when ST data is sparse or unavailable.
> >
> > **(3) Image-to-ST Sentence Retrieval**
> >
> > *Purpose:* This task evaluates the model’s ability to align and maintain consistency between tissue morphology and higher-level biological functions [3].
> >
> > *Biological Interpretation:* Each gene set represents a distinct biological process or phenotype. Successful retrieval indicates that the model effectively learns biologically interpretable visual–molecular relationships and can reliably associate histological regions with their relevant functional gene sets.
> >
> > [1] Maynard K R, Collado-Torres L, Weber L M, et al. Transcriptome-scale spatial gene expression in the human dorsolateral prefrontal cortex[J]. Nature neuroscience, 2021, 24(3): 425-436.
> >
> > [2] He B, Bergenstråhle L, Stenbeck L, et al. Integrating spatial gene expression and breast tumour morphology via deep learning[J]. Nature biomedical engineering, 2020, 4(8): 827-834.
> >
> > [3] Chen W, Zhang P, Tran T N, et al. A visual–omics foundation model to bridge histopathology with spatial transcriptomics[J]. Nature Methods, 2025: 1-15.

---

> ### Author Response · Authors · 2025-11-25
>
> **4. Transformations on gene expression:** We appreciate the reviewer’s comments regarding comparison fairness. Following the practices in BLEEP [1] and STEM [2], we apply a log1p transformation to the raw gene expression values to mitigate the effect of long-tailed expression distributions. Importantly, **we use the same preprocessing strategy across all benchmarked methods, ensuring consistency and fairness in comparison.** We clarified this in Section A.2 Gene Selection and Preprocessing (Line 795).
>
> > [1] Xie R, Pang K, Chung S, et al. Spatially resolved gene expression prediction from histology images via bi-modal contrastive learning[J]. Advances in Neural Information Processing Systems, 2023, 36: 70626-70637.
> >
> > [2] Zhu S, Zhu Y, Tao M, et al. Diffusion Generative Modeling for Spatially Resolved Gene Expression Inference from Histology Images[C]//The Thirteenth International Conference on Learning Representations.
>
> **5. Random Seeds:** We appreciate the reviewer's suggestion to test with random seeds. To this end, we trained our model on one HER2 fold (validated on samples D & F) using five different seeds (42, 123, 999, 2023, 555) for gene expression prediction task, obtaining an MSE/PCC of 0.7590 ± 0.091 / 0.3291 ± 0.054. These results are consistent with **our 4-fold cross-validation performance** (MSE/PCC = 0.8298 ± 0.1152 / 0.3172 ± 0.0891), indicating that the model maintains stable performance across different random initializations.
>
> **6. Non-smoothing clustering performance:** Following the reviewer’s request, we report the spatial clustering results without the spatial label-smoothing mechanism in the Table 6 and below. Without smoothing, our model still outperforms all competing methods, demonstrating that HiBio-ST inherently captures spatial information and does not rely on additional spatial regularization. All comparisons are conducted under the same experimental settings to ensure a fair and consistent evaluation.
>
> > | Model           | ARI    | NMI    |
> > | --------------- | ------ | ------ |
> > | Omiclip         | 0.1220 | 0.1747 |
> > | CONCH           | 0.1346 | 0.2070 |
> > | UNI             | 0.1322 | 0.2127 |
> > | ST-Path         | 0.1695 | 0.2568 |
> > | HiBio-ST (Ours) | 0.2503 | 0.3542 |
>
> **7. Zero-shot Clustering:** We thank the reviewer for raising this important point. Tools such as GraphST and STAGATE are indeed strong performers for spatial domain detection; however, these methods inherently rely on expensive spatial transcriptomics expression matrices. In contrast, our work focuses on a purely vision-driven formulation that investigates whether meaningful tissue domains can be inferred without using any ST sequencing data. Under this task formulation, we fairly compare different visual foundation models by evaluating their representation ability in a purely image-based zero-shot clustering setting.
>
> **8. Multi-stage architecture:** We appreciate the reviewer's comments on model design. In our ablation study, we have already evaluated the simplified single-scale InfoNCE setting suggested by the reviewer, corresponding to w.o Pathway and w.o TF-IDF in the results Table 4 and Table 5. This ablation removes the regional-level alignment and represents a single-scale contrastive model. The results reported under 4-fold cross-validation show that this single-scale variant consistently underperforms the full model across all downstream tasks, indicating that a single-scale design is insufficient to capture the multi-scale biological structure present in histology and ST data.
>
> Moreover, following the suggestion, we incorporated hyperparameter sensitivity analyses, with additional results reported in Tables 4, 5, and 7, and corresponding discussions added in *Section Pathway Overlap Ratio* (Line 481) and *Section Loss Weights Analysis (Line 490)*. Although different configurations lead to slight variations in performance, **introducing the proposed functional blocks consistently improves the model’s overall effectiveness.** In addition, the hierarchical alignment modules incur no noticeable increase in the number of model parameters, further demonstrating their practicality and efficiency.
>
> Lastly, our hierarchical design is inspired by the natural multi-scale organization of spatial transcriptomics, where local spot features reflect molecular programs within each microenvironment, spatially adjacent spots form coherent regional tissue domains, and biological pathways integrate these regions into global functional processes. Together, this structure enables the model to progressively link local molecular signals to regional architecture and ultimately to biologically meaningful tissue-level functions.

---

### Author Response · Authors · 2025-11-25

We sincerely thank all the reviewers and ACs for their constructive feedback. All revisions have been incorporated into the manuscript and highlighted in blue. Detailed point-by-point responses are provided below each reviewer’s comment.

---

### Author Response · Authors · 2025-12-02
**Summary to AC**

Dear AC,

We sincerely appreciate your time and navigating an especially challenging review process this year, given the unique circumstances. We appreciate the opportunity provided by the conference to submit a final message to assist your assessment.

**Reviewer outcomes and discussion highlights:**

**Post-rebuttal discussion showed substantial improvement in reviewer scores as evidenced by discussions below**:

- Reviewer 8X2p raised score from 4 to 6 after we addressed all concerns with additional experiments.
- Reviewer xsAL maintained orginal score 6 with positive feedback and acknowledged the improvements in our manuscript.
- Reviewer hemv recognized our efforts with active and positive discussion, and hasn't received the final response since addressing the second batch of concerns due to the current circumstances.
- Reviewer d1zK and RMdn didn't yet respond to our rebuttal.

Below is a summary of our paper and key rebuttal experiments that influenced reviewer scores.

**Paper summary:** We propose a biologically informed alignment framework that **reweights gene representations with TF–IDF and anchors them to curated pathways, together with a spatial-aware modeling strategy** that aggregates neighboring spots into coherent meso-scale units. HiBio-ST emerges as a multimodal foundation model with strong transferability, robust cross-dataset generalization, and superior performance across diverse tasks.

**Strengths highlighted by reviewers:**

- Unique incorporation of spatial and biological priors for visual-omics foundation model - d1zK, xsAL, RMdn, 8X2p
- Novel TF-IDF reweighting strategy - d1zK, RMdn, 8X2p
- Comprehensive experiments and ablations - d1zK, xsAL, RMdn, 8X2p
- Strong performance across tasks - d1zK, RMdn,

**Key concerns and rebuttal resolutions:**

##### 1. Pathway selection and integration (d1zK, xsAL, RMdn, 8X2p)

Pathways integration is automatic and driven solely by spot-specific gene sentences. Comparisons under pathway overlap ratio $\tau$ are summarized in Tables 4, 5, and 7, with analysis from Line 481.

> Low $\tau$ harms performance due to noisy pathway matches, while high $\tau$ offers limited anchors, causing performance to plateau or slightly decline.

##### 2. Modality-specific Foundation Models (xsAL, RMdn, hemv, 8X2p):

We compared with UMPIRE and STPath, which adopt pretrained CONCH and GigaPath as vision encoders, summarized in Figure 2 and Table 1.

> Thorough training process drives the model toward a new cross-modal optimum rather than preserving the pretrained structure.

##### 3. Clarification for downstream tasks (d1zK, xsAL, RMdn, hemv)

Our work emphasizes the biological utility of histology-derived representations to reduce reliance on costly ST profiling, with clarification provided from Line 966.

##### 4. Hyperparameter Sensitivity Analysis (d1zK, RMdn, 8X2p)

We incorporated hyperparameter sensitivity analyses, with results reported in Tables 4, 5, and 7, and discussions added in *Pathway Overlap Ratio* (Line 481) and *Loss Weights Analysis (Line 490)*.

> A moderate configuration yields the best trade-off between eliminating noisy and retaining informative anchors.

##### 5. Comparison with HVG/SVG selection (xsAL, hemv, 8X2p)

We included gene sentences using the HVG selection strategy under the same experimental settings, summarized in Table 4, 5, 7, and Figure 7, with analysis provided from Line 1294.

> HVG selection relies on slide-level statistics, providing only bulk-level information and failing to capture spot-level biological semantics.

##### 6. Spatial clustering (d1zK, 8X2p, RMdn)

We added zero-shot clustering on HER2 (Table 1 & Figure 8) and raw cluster performance (Table 6), along with region-identifier evolution (Figure 9) that explains how our model achieves stronger spatial structural representation.

> Combined signals of ST gene similarity and visual feature proximity guide nearby spots toward consistent regional identities, forming coherent tissue structures.

##### 7. Length of Gene Sentence (RMdn, xsAL)

We conducted an ablation on length of gene selection (Table 4), with discussion provided in Line 514.

> Performance improves with longer sentences to 250 tokens, after which added noise slightly reduces accuracy. Moderate token lengths strike the best balance between biological richness and model robustness.

##### 8. High-resolution performance and parameter size (hemv)

We added experiments on Visium HD datasets at high resolution, with results summarized in Tables 8 and 9 and detailed from Line 1325. Comparison of parameter sizes and architectures is summarized in Table 1.

Overall, our rebuttal and new analyses addressed reviewer concerns, strengthened validation, and reinforced HiBio-ST’s practical utility.

**We kindly ask the AC to consider our thorough revisions and new experiments, which earned praise and score increases from all participating reviewers.**

Thank you for your consideration!

Sincerely,

Authors

---

### Meta-Review · Area_Chair_Y7Uw · 2026-01-07

**Summary:**

### Summary
The paper proposes HiBio-ST, a visual-omics foundation model that aligns histology image representations with spatial transcriptomics signals using (i) TF-IDF-based gene selection to construct gene “sentences”, (ii) pathway-guided biological priors via curated gene sets, and (iii) a region-aware hierarchical alignment mechanism to capture meso-scale spatial structure. The method is evaluated on multiple downstream tasks including zero-shot spatial clustering, few-shot/gene expression prediction, and image-to-ST sentence retrieval, with ablations for key components.

### Strengths
- Integrates spatial information and pathway-level biological priors into a unified pretraining framework for histology-to-ST alignment.
- TF-IDF gene reweighting is biologically motivated (emphasizing locally enriched genes and reducing housekeeping-gene dominance).
- Comprehensive experimental suite across several downstream tasks with component-wise ablations and added sensitivity analyses in the rebuttal.
- Clear presentation and strong overall writing quality noted by multiple reviewers.

### Weaknesses
- Core empirical motivation is not fully supported: zero-shot clustering qualitative/biological fidelity is weak (e.g., unclear laminar structures on DLPFC), raising doubts about biological meaningfulness of learned embeddings.
- Limited novelty / stacking concern: method combines multiple existing ingredients (gene ranking, pathway priors, clustering/contrastive alignment) and the incremental benefit of each part is not always clearly isolated.
- Comparisons are incomplete or mismatched to key claims: missing/insufficient baselines against strong task-specific spatial tools and alternative gene selection strategies (HVG/SVG) and/or recent multimodal baselines in the original submission.
- Generalizability and practicality questions (dataset coverage, high-resolution ST, stability across seeds/hyperparameters) required substantial clarification and additional experiments during rebuttal.

While the approach is well-motivated and the framework is carefully engineered, reviewers raise substantial concerns that the empirical evidence does not convincingly demonstrate biologically faithful spatial structure, especially in zero-shot settings. The contribution is perceived as incremental and the original comparisons/ablations were insufficient to isolate the necessity of the multi-stage design and the gene/pathway choices. Despite improvements in the rebuttal, key doubts remain about whether the method provides reliable, biologically meaningful gains beyond existing spatial transcriptomics analyses and competing foundation-model baselines.

**Reviewer Concerns:**

- d1zK
  - addressed: clarified w/o TF-IDF uses same gene count (top-expressed); added pathway-overlap sensitivity; clarified preprocessing (log1p) and seed stability; reported clustering without smoothing; argued why task-specific ST-only tools are not comparable for vision-only zero-shot.
  - still outstanding: biological fidelity of DLPFC spatial domains remains unconvincing; concern about over-stacking/necessity of hierarchical design only partially resolved; missing direct comparisons to strong ST task-specific methods for domain detection remains a practical gap for biological users.

- xsAL
  - addressed: clarified downstream task purpose/interpretation; expanded discussion on pathway prior stability and encoder choice; acknowledged graph-based spatial modeling as future direction; reviewer explicitly maintained rating after rebuttal.
  - still outstanding: limited discussion/experiments on alternative gene-ranking mechanisms beyond TF-IDF; remaining concerns are mostly positioning and breadth rather than a specific unaddressed bug.

- RMdn
  - note: reviewer indicated inability to submit a qualified review on time; no substantive review content to assess rebuttal impact.

- hemv
  - addressed: defined missing notation; clarified Eq. (11); added experiments on high-resolution Visium HD and added HVG baseline comparison per reviewer request.
  - still outstanding: even after added experiments, the reviewer’s concerns suggest the paper’s original generalizability claims required significant post-hoc additions; questions remain about broader multi-resolution pretraining and the strength of evidence for TF-IDF vs other established strategies (e.g., SVG) in a unified setting.

- 8X2p
  - addressed: added discussion/examples motivating TF-IDF; added UMPIRE/STPath comparisons; added additional results for HER2/DLPFC coverage; clarified pathway selection is automatic spot-level activation; responded to SVG-ranking concern with rationale; reviewer raised score to 6.
  - still outstanding: debate remains about whether comparisons to SVG-based strategies in pretraining are fully settled for the community; however, major reviewer concerns appear largely resolved.

**Reviewer Scores:**

- d1zK: no change
- xsAL: no change (explicitly maintained rating)
- RMdn: no change
- hemv: no change (raised significant remaining concerns; no explicit score increase)
- 8X2p: would increase (explicitly raised score to 6)

---

### Decision · Program_Chairs · 2026-01-26

Reject